# Patient-derived monoclonal antibody neutralizes HCV infection in vitro and vivo without generating escape mutants

Hiroshi Yokokawa[1,2☯]*, Midori Shinohara[3☯], Yuji Teraoka[4,5‡], Michio Imamura[4,5‡], Noriko Nakamura[1☯], Noriyuki Watanabe[2], Tomoko Date[2], Hideki Aizaki[2], Tomokatsu Iwamura[1], Hideki Narumi[1], Kazuaki Chayama[5,6,7], Takaji Wakita[2]*

1 Pharmaceutical Research Laboratory, Toray Industries, Inc., Kanagawa, Japan, 2 Department of Virology II, National Institute of Infectious Diseases, Tokyo, Japan, 3 Medical & Biological Laboratories Co., Ltd., Tokyo, Japan, 4 Department of Gastroenterology and Metabolism, Graduate School of Biomedical and Health Science, Hiroshima University, Hiroshima, Japan, 5 Research Center for Hepatology and Gastroenterology, Hiroshima University, Hiroshima, Japan, 6 Collaborative Research Laboratory of Medical Innovation, Graduate School of Biomedical and Health Sciences, Hiroshima University, Hiroshima, Japan, 7 RIKEN Center for Integrative Medical Sciences, Yokohama, Japan

☯ These authors contributed equally to this work.
‡ YT and MI also contributed equally to this work.
* hiro_ykkw@msn.com (HY); wakita@nih.go.jp (TW)

**Data Availability Statement:** All relevant data are within the paper and its Supporting Information files.

## Abstract

In recent years, new direct-acting antivirals for hepatitis C virus (HCV) have been approved, but hepatitis C continues to pose a threat to human health. It is important to develop neutralizing anti-HCV antibodies to prevent medical and accidental infection, such as might occur via liver transplantation of chronic HCV patients and needle-stick accidents in the clinic. In this study, we sought to obtain anti-HCV antibodies using phage display screening. Phages displaying human hepatocellular carcinoma patient-derived antibodies were screened by 4 rounds of biopanning with genotype-1b and -2a HCV envelope E2 protein adsorbed to magnetic beads. The three antibodies obtained from this screen had reactivity against E2 proteins derived from both genotype-1b and -2a strains. However, in epitope analysis, these antibodies did not recognize linear peptides from an overlapping E2 epitope peptide library, and did not bind to denatured E2 protein. In addition, these antibodies showed cross-genotypic neutralizing activity against genotype-1a, -1b, -2a, and -3a cell culture-generated infectious HCV particles (HCVcc). Moreover, emergence of viral escape mutants was not observed after repeated rounds of passaging of HCV-infected cells in the presence of one such antibody, e2d066. Furthermore, injection of the e2d066 antibody into human hepatocyte-transplanted immunodeficient mice inhibited infection by J6/JFH-1 HCVcc. In conclusion, we identified conformational epitope-recognizing, cross-genotypic neutralizing antibodies using phage display screening. Notably, e2d066 antibody did not select for escape mutant emergence in vitro and demonstrated neutralizing activity in vivo. Our results suggested that these antibodies may serve as prophylactic and therapeutic agents.

**Funding:** This study was supported by The Japan Society for the Promotion of Science KAKENHI (No. JP207K08369) and The Japan Agency for Medical Research and Development AMED (Nos. JP21fk02100539j0103, 21fk0310103j0305, 21fk0210065h0002, and 21fk0210065j1002). This work also was supported by a grant for Research on Health Sciences Focusing on Drug Innovation from the Japan Health Sciences Foundation (No. KHC1213). The funders had no role in study design, data collection and analysis, decision to publish, or preparation of the manuscript.

**Competing interests:** Hiroshi Yokokawa, Noriko Nakamura, Tomokatsu Iwamura, and Hideki Narumi are employees of Toray Industries, Inc. Midori Shinohara is an employee of Medical & Biological Laboratories Co., Ltd. The researchers were employed by these commercial funders and received funding for patent rights to this research. This study was patent filed in 2015 (WO2015/141826), but the rights application was withdrawn in 2018. The researchers and commercial funders do not benefit from the submission of this study to PLOS ONE. This does not alter our adherence to PLOS ONE policies on sharing data and materials.

## Introduction

Hepatitis C virus (HCV) is an RNA virus belonging to the family *Flaviviridae*, genus *Hepacivirus*, and originally was identified as a major causative virus of non-A, non-B hepatitis [1]. HCV infects primarily through blood transfusion, but in recent years the number of patients newly infected with HCV through blood transfusion has been greatly reduced following implementation of a highly sensitive HCV blood screening method. Nonetheless, there are still infection risks for HCV, such as injecting drugs through reuse or sharing of needles and syringes. The World Health Organization has reported that there are 58 million people with chronic HCV infection worldwide, and 1.5 million new infections are observed annually. In 2019, approximately 290,000 people died by this infectious disease [2]. Historically, interferon-based therapies have been used to eradicate HCV, but the efficacies of these treatment were not satisfactory. Recently, direct-acting antivirals (DAAs) have been developed; these therapies have been reported to provide efficacy in more than 95% of patients with HCV. However, at present, no prophylactic agents, such as neutralizing antibodies and vaccines, are available against HCV infection. The development of a prophylactic agent against HCV is necessary to prevent new infection and re-infection after DAA treatment. HCV envelope proteins are believed to be responsible for the binding of HCV to the cell surface, the first step in HCV infection. In an effort to develop modalities capable of preventing HCV infection, researchers have investigated patient sera for antibodies against the HCV envelope proteins. It has been reported that approximately 20–30% of patients with acute HCV infection are able to clear the virus spontaneously at 1 year after infection [3, 4]. It also has been reported that neutralizing antibodies are induced early in patients with spontaneous clearance, but appear later in persistently infected patients [5]. Other work has examined the efficacy of preparations containing a mixture of immunoglobulins obtained from the blood plasma of multiple patients with chronic hepatitis C who are positive for anti-HCV antibodies. However, mixed preparations of patient-derived anti-HCV antibody do not reduce the amount of HCV in plasma, even if administered concomitantly with liver transplantation [6]. Other cell-based antibody production techniques also have been investigated, including antibodies obtained from hybridomas generated by fusion of B cells from a patient with hepatitis C to cells from viral protein-inoculated animals. For example, MBL-HCV1 [7], a human monoclonal antibody isolated using a human immunoglobulin transgenic mouse inoculated with E2 envelope glycoprotein, was shown to have cross-genotypic neutralizing activity against HCV pseudoparticle (HCVpp) infection. Interestingly, pretreatment of chimpanzees with MBL-HCV1 antibody prevented acute HCV infection [8]. In a chimpanzee model of chronic HCV infection, treatment with MBL-HCV1 reduced plasma viral load to below the limit of detection, where the viral level remained for 14 days. However, HCV reemerged in the treated chimpanzees by 21 days after infusion of MBL-HCV1. In such cases, resurgence has been postulated to reflect acquisition of resistance to the neutralizing activity of MBL-HCV1 through viral genome mutation. Specifically, when an anti-HCV antibody is used as treatment, there is concern about the emergence of "escape mutants". In the present study, we cloned antibody genes from chronic hepatitis C patients, and established a single-chain variable fragment (scFv) -phage display library screening system to identify new HCV-specific antibodies. We describe the characterization of the resulting antibodies based on their binding affinity for the E2 glycoprotein, cross-genotypic neutralizing activity, escape mutant suppressive properties, and neutralizing activity in vivo.

## Materials and methods

### Cell culture

Huh-7.5.1 cells (a generous gift from Dr. Francis V. Chisari, The Scripps Research Institute) and COS1, HEK293 cells (American Type Culture Collection, Manassas, VA) were cultured in

Dulbecco's Minimal Essential Medium (DMEM) containing 10% fetal bovine serum (FBS). CHO-K1 cells (American Type Culture Collection) were cultured in Ham's F-12 nutrient mixture containing 10% FBS. All mammalian cell cultures were maintained in a 5% $CO_2$ environment at 37°C, as described previously [9, 10].

### Preparation of antibody phage library

The antibody phage library was prepared as described previously [11, 12]. Briefly, messenger RNAs encoding antibodies were isolated from human splenocytes derived from the spleens of 4 patients with HCV-positive hepatocellular carcinoma. The pooled mRNAs were used for synthesizing cDNA by RT-PCR. Sequences encoding the variable regions of the heavy chain (VH) and those encoding the variable and constant regions of the light chain (VLCL) were amplified (separately) from the cDNAs by PCR using mixed primers (S2 Table). The amplified DNA sequences encoding the VH and VLCL regions were inserted into the pTZ19R phagemid vector, which contains a linker DNA encoding a linker peptide sequence such as $(GGGGS)_3$. The resulting constructs were constituted of a single-chain variable fragment (scFv) gene library in which the sequences encoding the VH region, the linker, the VL region, and the CL region are assembled in that order. This scFv gene library was transformed into Escherichia coli DH12S™ (Thermo Fisher Scientific, Waltham, MA). The bacterial library then was super-infected with a helper phage (M13K07, Thermo Fischer Scientific) to generate a scFv phage library. The phages of the resulting library display scFv molecules in which the VH and the VLCL regions derived from human antibodies are connected via the linker sequence. This study was approved by the Medical & Biological Laboratories Co., Ltd Ethics Committee (Ethical Review No.067) and was implemented according to the Ethical Guidelines for Human Genome/Gene Research enacted by the Japanese Government and the Helsinki Declaration.

### Recombinant proteins

To obtain the His-tagged E2 proteins, DNA sequences encoding the E2 protein of the TH strain (genotype 1b; amino acid (aa) 384–711) [13] or of the JFH-1 strain (genotype 2a; aa 384–714) [14] were cloned into pMT/BiP/V5-His, a S2 cell expression vector (Thermo Fisher Scientific). The resulting constructs were transfected (separately) into S2 cells (Thermo Fisher Scientific) using the calcium phosphate precipitation method (Calcium Phosphate Transfection Kit, Thermo Fisher Scientific); cells then were cultured in Schneider's Drosophila Medium supplemented with 10% fetal bovine serum (FBS), penicillin (50 units/mL), and streptomycin (50 μg/mL) at 28°C. After 24 hours, the medium was replaced with Schneider's Drosophila Medium supplemented with 10% FBS, penicillin (50 units/mL), streptomycin (50 μg/mL), and hygromycin B (300 μg/mL); subculturing at 28°C was continued for approximately three weeks by exchanging the medium every 3 or 4 days, thereby establishing a stable cell strain. For induction of the expression of the E2 protein, CuSO4 (at a final concentration of 0.5 mmol/L) was added to a culture of 2 x 106 S2 cells (in 40 mL per 225-cm2 flask) and the cells were incubated for 5 days at 28°C. The spent culture medium was collected and centrifuged (500 x g, 5 minutes) to remove cells and obtain the culture supernatant. The culture supernatant was subjected to chromatography using Ni-NTA agarose, permitting purification of the His-tagged E2 protein of the HCV TH strain (genotype 1b) or of the HCV JFH-1 (genotype 2a).

To obtain the E2 protein-Fc fusion proteins, DNA sequences encoding the E2 protein of the TH strain (genotype 1b; aa 384–711) or of the J6CF strain (genotype 2a; aa 384–720) [15] were cloned (separately) into pCR-TOPO (Thermo Fisher Scientific). The gene fragments encoding the respective E2 proteins then were subcloned (separately) into p3xFLAG-CMV-13

(Sigma-Aldrich, St. Louis, MO), such that the E2 proteins were encoded with N-terminal signal peptides. The DNA fragment encoding the signal peptide-E2 fusion proteins then were further subcloned (separately) into the CDM-mIL7R-Ig vector. This vector harbors sequences encoding a chimeric protein consisting of a mouse IL-7 receptor and a human immunoglobulin Fc region; the resulting clones therefore encoded an E2 protein (the antigen) attached to the Fc domain of human immunoglobulin, a fusion protein referred to as the TH-Fc or J6CF-Fc protein. The resulting constructs (CDM-TH-Fc and CDM-J6CF-Fc, respectively) were transfected into COS1 cells by the Diethylaminoethyl (DEAE)-dextran method (DEAE-dextran transfection kit, Sigma-Aldrich) to permit expression of the E2-Fc proteins. Cells then were cultured in DMEM containing 10% FBS, penicillin (50 units/mL), and streptomycin (50 μg/mL). The resulting E2-Fc proteins were purified from the cell culture supernatants by chromatography using Prosep-A (Merck Millipore, Billerica, MA), a Protein-A-bound support.

## Screening of antibody-display phages for ability to bind to HCV E2 protein

The screening of antibody phages was performed using a method described previously [10, 11]; the technique is provided briefly here. Each HCV E2 protein (from the TH and JFH-1 strains) was allowed to bind to the surface of a magnetic bead. The antibody phage library was added to the E2 protein-bead mixture and allowed to bind, and the beads then were washed with phosphate-buffered saline (PBS) containing 0.1% Tween 20 to remove non-specifically bound phages. Specifically bound phages were eluted from the magnetic beads using 0.1 mol/L glycine-HCl (pH 2.2); the eluted phages then were allowed to infect *E. coli* and amplified. These steps were repeated for a total of 4 rounds using (in order) JFH-1 E2-His/TALON-beads, JFH-1 E2-His/TALON-beads, JFH-1 E2-His/anti-His/Protein-G-beads, and TH E2-His/TALON-beads, thereby enriching for phages with specific E2 binding. The nucleotide sequences of the regions encoding the heavy- and light-chain variable domains ($V_H$ and $V_L$, respectively) in the scFv displayed by the specifically bound phages were analyzed by standard techniques.

## Preparation of scFv antibodies

DNA from each of the phagemids (selected based on the encoded antibody) was digested with restriction enzyme SalI, converted into a plasmid encoding a Protein A-fused antibody (scFv-PP antibody) through self-ligation, and transformed into *E. coli* DH12S. The transformants were cultured overnight at 30˚C in 25 mL of 2x YTGA (YT medium supplemented with 1% glucose and 100 μg/mL ampicillin). An aliquot (10 mL) of the overnight culture was used to inoculate 1 L of 2x YTA (YT medium supplemented with 100 μg/mL ampicillin) and grown for 3 hours at 30˚C. Isopropyl β-D-1-thiogalactopyranoside (IPTG; 1 mL of 1 mol/L) was added and culturing at 30˚C was continued for another 20 hours. The resulting culture was centrifuged at 4˚C and 10,000 x g for 10 minutes to pellet the cells. Ammonium sulfate (313 g) was slowly added to the cleared supernatant, and the mixture was stirred at room temperature for 30 minutes before being centrifuged at 4˚C and 10,000 x g for 30 minutes. The resulting precipitate was resuspended in PBS containing 20 mL of Complete Protease Inhibitor Cocktail (Roche Diagnostics, Basel, Switzerland). This solution was dialyzed against PBS overnight at 4˚C, followed by centrifugation at 4˚C and 12,000 x g for 10 minutes. The resulting supernatant was passed (via natural dripping) at room temperature over a column filled with 2 mL of Immunoglobulin G (IgG) Sepharose (GE Healthcare, Little Chalfont, UK). After the column was washed with 300 mL of PBS, the antibody was eluted with 8 mL of 0.2 mol/L glycine (pH 3.0). The eluate was adjusted to pH 7.0 with 2 mol/L Tris buffer, transferred to a fresh tube,

and concentrated by dialysis against PBS using an Amicon Ultra-15 Centrifugal Filter unit (Merck Millipore). An aliquot of the final product was subjected to sodium dodecyl sulfate polyacrylamide gel electrophoresis (SDS-PAGE) and the protein concentration was determined by comparison to a standard.

## Preparation of immunoglobulin G (IgG) antibody

PCR was performed using the $V_H$- and $V_L C_L$ region-encoding sequences of the scFv antibody-encoding clones as templates and amplification primers specific for the H- and L-chain-encoding sequences. The amplified $V_H$-encoding product was ligated into a vector encoding a human IgG1 constant region, and the amplified $V_L C_L$-encoding product was ligated into an L-chain construction vector; the two resulting constructs were linked to each other to obtain a plasmid DNA (Mammalian PowerExpress System, TOYOBO, Osaka, Japan) containing a complete IgG antibody-encoding gene. The plasmid DNA was digested with a restriction enzyme and linearized, and 40 μg of the linearized plasmid was introduced into $1 \times 10^7$ CHO-K1 cells (American Type Culture Collection) by electroporation at 250 V and 800 μF. Immediately after electroporation, the cells were suspended in Ham's F12 medium (FUJIFILM Wako Pure Chemical, Osaka, Japan) containing 10% FBS and cultured at 37°C in a 5% $CO_2$ incubator. After 24 hours, selection was initiated by addition of puromycin (Sigma-Aldrich) to the culture to a final concentration of 10 μg/mL. After an interval of 10 to 14 days, the cells were washed with 20 mL of PBS, treated with 1 mL of 0.05% Trypsin-EDTA (FUJIFILM Wako Pure Chemical), detached from the plate into 5 mL of Ham's F12 medium, and the number of cells was counted. Based on the determined cell density, limiting dilution was performed to yield 0.2 cell/200 μL/well (across five 96-well plates). After culturing for 14 days, cells that exhibited strong expression of an IgG antibody were identified by screening the culture supernatant of each well using an enzyme immune assay (EIA). IgG-producing cells were conditioned in a serum-free medium (EX-CELL CD CHO Fusion; Nichirei Biosciences, Tokyo, Japan); the cells then were cultured in 6-well plate (2 mL/well) at an initial cell count of $2 \times 10^5$ for 10 days. The resulting supernatant was passaged over a column filled with 1 mL of rProtein A Sepharose Fast Flow (GE Healthcare) at a flow rate of 1 drop/2 seconds to allow the expressed protein (IgG) to bind to the column. Non-adsorbed components were removed by washing with 10 mL of PBS at a flow rate of 1 drop/2 seconds, and the IgG was eluted with 10 mL of elution buffer (0.2 mol/L glycine, pH 3) at a flow rate of 1 drop/second. The resulting eluate was concentrated by ultrafiltration into 1 mL PBS using an Amicon Ultra-15 Centrifugal Filter unit (Merck Millipore). The antibody protein was quantified by SDS-PAGE as above.

## HCV pseudoparticle (HCVpp) production

Murine leukemia virus pseudoparticles were generated using the method described previously [16]. 293T cells (American Type Culture Collection) were seeded in 10-cm dishes at $2 \times 10^6$ cells/dish and were cultured in 8 mL DMEM with 10% FBS at 5% $CO_2$. On the following day, each dish of these cultures was transfected, using FuGENE6 transfection reagent (Roche Diagnostics), with a mixture consisting of plasmid DNAs corresponding to the *gag-pol* packaging vector (3.1 μg), the transfer vector (3.1 μg), and the HCV glycoprotein-encoding vector (1 μg). The HCV glycoprotein-encoding vector harbored DNA sequences encoding a segment of the HCV glycoprotein corresponding to amino acid (aa) 132–747 from the TH strain (genotype 1b), or aa 132–750 from the J6CF strain (genotype 2a). After 48 hours, pseudoparticle-containing supernatants were collected and passed through a 0.45-μm-pore-size filter (Corning, NY, USA). The collected pseudoparticles were stored at -80°C until use.

## Chimeric cell culture-generated HCV particles (HCVcc) preparation

Chimeric HCVcc (H77/JFH-1 (genotype (GT) 1a), TH/JFH-1 (GT1b), J6/JFH-1 (GT2a), and S310/JFH-1 (GT3a)) were generated as described previously [17]. Chimeric HCV plasmids (pJ6/JFH-1, pH77/JFH-1, pTH/JFH-1, and pS310/JFH-1; genotypes 2a, 1a, 1b, and 3a, respectively) were generated by replacing the region spanning the 5' UTR to p7 of pJFH-1 with that of the respective strain. Full-length HCV RNA synthesis and transfection into Huh-7.5.1 cells were performed as described previously [18]. Cell culture supernatants were collected 72 hours after transfection and used as a seed virus stock. For large-scale culture, Huh7 cells were infected with this seed virus stock, and the infected cells were expanded into CellSTACK (Corning) with 2% FBS-supplemented medium. Supernatant was collected and passed through a 0.45-μm filter. The supernatant was concentrated by hollow fiber ultrafiltration using a UFP-500-C-8MA (GE Healthcare), and diafiltered against PBS at 5 times the initial volume. The resulting concentrated supernatant was purified by ultra-centrifugation over a 20%/60% sucrose cushion as described previously [19]. Fractions (1 mL each) were collected from the bottom of the centrifuge tube, and the densities and protein concentrations of each fraction were measured. Infectious titers and HCV core protein concentrations also were measured as described previously [19]. Purified HCVcc was prepared by collecting the peaks of infectivity, and was concentrated using an Amicon Ultra-15 unit (100,000 molecular weight cut-off; Merck Millipore).

## Enzyme immune assay

The anti-E2 antibodies were assayed for their comparative affinity for the J6CF E2-Fc and TH E2-Fc proteins. All steps were performed at room temperature. A sample of each of the various monoclonal antibodies was subjected to either a 3- or 10-fold serial dilution, and the dilutions were distributed to an EIA plate (Thermo Fisher Scientific) in which the J6CF E2-Fc or TH E2-Fc protein had been immobilized (0.5 μg/well). An antibody, HR1-007 raised against *Protobothrops flavoviridis* venom hemorrhagic factor was employed as a control human IgG. After incubation for 1 hour, the plates were washed with PBS containing 0.05% Tween 20 (PBS-T), and horseradish peroxidase (HRP) -conjugated goat anti-human IgG F(ab')2 (1:5000; Thermo Fisher Scientific) was added to each well. After incubation for 1 hour, the plates were washed with PBS-T, and color was developed with ELISA POD substrate TMB solution (Nacalai Tesque, Kyoto, Japan); reactions were quenched with 1 mol/L sulfuric acid, and absorbance (at 450 nm) was measured using a microplate reader.

## HCVpp inhibition assay

Evaluation of neutralizing activity of anti-E2 antibody against HCVpp infection was performed as described previously [14]. HCVpp, which harbor a luciferase-encoding construct, were mixed with IgG or PBS for 30 minutes at room temperature and then used to infect naïve Huh-7.5.1 cells. Monoclonal anti-CD81 antibody (JS-81, BD Pharmingen, Franklin lakes, NJ) was used as positive control in this assay. The infected cells were harvested at 72 hours after infection and lysed with Cell Culture Lysis Reagent (Promega, Madison, WI). Luciferase activity was quantified using the Luciferase Assay System (Promega). Neutralizing activity was calculated and represented as the % neutralization by comparison with the luciferase activities of the well inoculated with PBS-HCVpp.

## IgG biotinylation

Each of the antibodies (the e2d066 IgG, the e2d073 IgG, the e2d081 IgG, the AR3A antibody, and the MBL-HCV1 antibody) was biotinylated. The biotinylation was performed using

EZ-Link® Sulfo-NHS-LC-Biotin (Thermo Fisher Scientific) according to the manufacturer's instructions. Briefly, 2.4 μL of 20 mmol/L Sulfo-NHS-LC-Biotin was added to 100 μL of each IgG antibody (diluted with PBS to 0.1 mg/mL), and the mixtures were held on ice for 2 hours. Next, the mixture was desalted by passage over a Zeba® Spin Desalting Column (Thermo Fisher Scientific), removing the unreacted biotin and yielding the biotinylated antibody.

## Denatured and naïve E2 protein EIA using biotinylated IgG

The TH E2-Fc protein was denatured by heat treatment at 95˚C for 3 minutes in 50 mmol/L Tris-HCl (pH 7.0) containing 2% SDS and 5% 2-mercaptoethanol. The native (non-denatured) TH E2-Fc protein and the denatured TH E2-Fc protein were diluted (separately) with PBS to a concentration of 0.5 μg/mL, and 50 μL of the dilution was added to each well of an immuno-plate (Thermo Fisher Scientific), which was held at 4˚C overnight to immobilize the protein on the plate. The protein solution was removed, and the plates were blocked by addition of 200 μL/well of Blocking One solution (Nacalai Tesque) prepared according to the manufacturer's instructions. Next, the biotinylated antibody was subjected to a 3-fold serial dilution with PBS and distributed to the reaction plates at 50 μL/well. The plates were held at room temperature for 1 hour and then washed with PBS-T; an aliquot (50 μL) of HRP-conjugated avidin (1:300 in PBS-T; GE Healthcare) was added to each well. After incubation at room temperature for 1 hour, the plates were washed with PBS-T, and color development, quenching, and plate reading were performed as described above for EIA.

## Epitope analysis with overlapping epitopes

With respect to the amino acid sequence corresponding to the E2 protein of the TH strain (i.e., the amino acid sequence corresponding to aa 384 to 717, defining the initiating methionine at the N-terminus of the precursor protein of the TH strain was position 1), a series of overlapping 12-mer peptides (designated peptides No. 1 to 82) were designed; each represented a shift of 3 amino acids toward the C-terminus, starting from the N-terminus of the full-length protein. Each peptide was biotinylated at its N-terminus, and had a glycine amide appended at the C-terminus. A stock solution of each peptide was generated by dissolving in dimethyl sulfoxide (DMSO), and the stock solutions then were diluted in PBS to a concentration of 0.01 mmol/L. The diluted peptide solutions were distributed to a streptavidin-coated plate (Thermo Fisher Scientific) at 50 μL/well, and the plate was held at room temperature for 2 hours to allow the peptides to adsorb to the plate. The peptide solution was removed, and plates were blocked by addition of 200 μL/well of Blocking One solution (Nacalai Tesque) prepared according to the manufacturer's instructions. The plates were held overnight at 4˚C for blocking. Following removal of the blocking solution, the plates were washed four times with PBS-T, and antibodies (diluted to 1 μg/mL with PBS-T) were added at 50 μL/well; the reactions were allowed to proceed at room temperature for 1.5 hours. Following removal of the antibody solution, the plates again were washed four times with PBS-T, and an HRP-conjugated anti-human IgG sheep antibody (1:5000 in PBS-T; GE Healthcare) was added at 50 μL/well. Following reaction at room temperature for 1 hour, the antibody solution was removed, and the plates were washed five times with PBS-T. Color then was developed using a peroxidase coloring kit (Type T: Sumitomo Bakelite Co., Ltd., Tokyo, Japan), and the absorbance was measured at 450 nm, thereby detecting antibody binding to the peptides.

## Competitive EIA

Antibodies were subjected to 3-fold serial dilution, starting from a concentration of 20 μg/mL. Equivalent amounts of biotinylated e2d066 IgG were added to each dilution; the mixtures then

were stirred and transferred to an EIA plate (Thermo Fisher Scientific) in which the TH E2-Fc protein had been immobilized (0.5 μg/well) at room temperature (all microplate incubations were performed at room temperature unless otherwise indicated). After 1 hour, the plate was washed with PBS-T, and avidin-HRP (1:3000; GE Healthcare) was added. After 1 hour, the plate was washed with PBS-T, and color development, quenching, and plate reading were performed as described above for EIA.

## HCVcc neutralizing assay

Chimeric HCV particles of each genotype (H77/JFH-1, TH/JFH-1, J6/JFH-1, and S310/JFH-1) were mixed with IgG or scFv at 1:1, and the mixtures were allowed to react at room temperature for 30 minutes. The antibody-HCVcc mixture was added to naïve Huh-7.5.1 cells (2 x 10^4 cells/well, 48 well-plate) at 100 μL/well (multiplicity of infection = 0.1), and the plate was incubated for 3 hours at 37˚C in a 5% $CO_2$ incubator. The antibody-HCVcc mixture was removed and replaced with 500 μL of DMEM containing 10% FBS, and the cells were cultured for 72 hours at 37˚C in a 5% $CO_2$ incubator. Monoclonal anti-CD81 antibody (JS-81, BD Pharmingen, Franklin lakes, NJ) was used as a positive control in this assay. After 72 hours of incubation, the cells were washed with PBS, and Passive Lysis Buffer (Promega) was added to the plate at 100 μL/well to generate a cell lysate. HCV core protein in the collected cell lysate was quantified using a Lumipulse Ortho HCV Ag (Ortho Clinical Diagnostics, Tokyo, Japan). Based on the molar concentration of the HCV core protein determined by this assay, the HCVcc infection neutralizing activity of each antibody (scFv or IgG antibody) was calculated by normalizing to the amount of HCV core protein in the PBS-treated HCVcc-infected cells.

## Selection of neutralizing antibody-resistant escape mutants

The evaluation of properties for suppressing escape mutant emergence was performed in accordance with a method described previously [20]. Briefly, each IgG antibody (diluted with PBS) was mixed with the J6/JFH-1 HCVcc and allowed to react at 37˚C for 1 hour. The antibody-HCVcc mixture then was added to Huh7 cells seeded in a 12-well plate, to permit the HCVcc to infect the cells. After three days, the cells were subcultured into a 6-well plate, and the culture supernatant was collected on the 3rd and 6th days of subculturing. The presence of J6/JFH-1 HCVcc in the collected culture supernatant was confirmed by performing an infectious titer measurement (as described below). The culture supernatant (collected on the 3rd or 6th day) then was used for subsequent infection. Specifically, the collected culture supernatant containing the J6/JFH-1 HCVcc produced through infection culture was mixed again with the respective IgG antibody, and the mixture was added to uninfected Huh-7.5.1 cells to permit HCVcc to infect the cells. These steps were repeated for a total of 8 times. The infectious titer of the J6/JFH-1 HCVcc contained in the culture supernatant collected from the final round of infection (referred to as the J6/JFH-1 EM_antibody, e.g. J6/JFH-1 EM_e2d066) was measured. Specifically the J6/JFH-1 EM_antibody was added to each well of naïve Huh-7.5.1-seeded 96-well plate and the plate was cultured at 37˚C for 72 hours. The cells then were washed with PBS and immobilized by treatment with methanol at -20˚C for 20 minutes. The immobilized cells were blocked by incubation for 1 hour with Block Ace solution (DS Pharma Biomedical, Tokyo, Japan), and the plate was washed with PBS. An anti-core monoclonal antibody 2H9 [9] was added to a concentration of 10 μg/mL, and the plate was incubated at room temperature for 1 hour. The supernatant was removed and the plate was washed, and Alexa Fluor 488-conjugated anti-mouse IgG (Thermo Fisher Scientific) was added; the plate then was incubated at room temperature for 1 hour. After a further wash, PBS was added at 50 μL/well, and the cells were observed with a fluorescence microscope (Leica TCS SPE; Leica Microsystems K.K.,

Tokyo, Japan). The number of fluorescent cells was counted and the infectious titer was shown as focus-forming units/mL (FFU/mL).

The infection-neutralizing activity of the IgG antibody against the J6/JFH-1 EM_antibody or against J6/JFH-1 EM_control (that is, a strain subjected to selection using the unrelated control antibody HR1-007; the comparative control), was measured to permit detection of escape mutants. First, the J6/JFH-1 EM_antibody or the J6/JFH-1 EM_control (both at 100 ffu) was combined with an equivalent volume of diluted IgG antibody, and the mixture was incubated at 37˚C for 1 hour. Separately, the culture supernatant was removed from Huh7 cells that had been seeded in each well of an 8-well chamber slide (Thermo Fisher Scientific) and replaced with 100 μL of the antibody-HCVcc mixture; the resulting cultures were incubated at 37˚C for 24 hours. The culture supernatant again was removed and this time replaced with 200 μL of DMEM containing 10% FBS and 1% MEM nonessential amino acid solution (Thermo Fisher Scientific); the resulting cultures were incubated at 37˚C for 48 hours. A fluorescently labeled antibody was allowed to act on the cells of each well in the same manner as in the measurement of the infectious titer as described above; the number of fluorescent cells then was counted and used to determine the infectious titer. The infectious titers were determined by the infection of J6/JFH-1 EM_antibody or the J6/JFH-1 EM_control into the cells in the presence of IgG antibody.

### HCVcc infection in uPA/SCID human liver chimeric mice

Human hepatocyte-transplanted urokinase-type plasminogen activator/severe combined immunodeficiency (uPA/SCID) mice were inoculated with J6/JFH-1 HCVcc on Days 0, 28, and 42 (with HCV RNA at copy numbers of $10^2$, $10^3$, and $10^3$ per mouse, respectively). Inoculated mice were treated (800 μg/body, intraperitoneal injection) with anti-E2 antibody e2d066 or control human IgG HR1-007 [12] at Day -1 (i.e., one day before) and Days 1, 4, 8, and 11 after the first (Day-0) challenge with J6/JFH-1 HCVcc. Blood samples of inoculated mice were collected weekly and processed to obtain serum samples. Serum HCV RNA levels were determined using real-time detection reverse transcription-polymerase chain reaction (RTD-PCR). Human albumin (h-Alb) in the serum samples also was monitored to assess the safety of the antibodies. All animal protocols described in this study were performed in accordance with the National Institutes of Health "Guide for the Care and Use of Laboratory Animals" and the guidelines of the local committee for animal experiments, and the experimental protocol was approved by the Ethics Review Committee for Animal Experimentation of the Graduate School of Biomedical and Health Sciences, Hiroshima University (A14-195).

### Statistical analysis

Statistical analysis was conducted by using EXSUS software version 10.1.4 (EPS Corporation, Tokyo, Japan). Statistical analysis was carried out using the one-way analysis of variance (ANOVA) with a Williams' test for multiple groups. A P value less than 0.025 was considered significant. Data are reported as means ± standard errors of mean (SEM) as indicated.

## Results

### Phage display screening

Prepared phages displaying human hepatocellular carcinoma patient-derived scFv were screened by 4 rounds of biopanning using HCV envelope E2 protein adsorbed to magnetic beads as the bait (S1 Fig). For the beads, we used E2 proteins of genotype 1b and genotype 2a, which are the most frequently infected genotypes in Japan. Because the mRNA samples used

**Table 1. Results of phage display screen.**

| | Antigen | Used amount of antigen/beads | Input phages (PFU[a]) | Wash (times) | Output phages (PFU[a]) | Recovery rate |
|---|---|---|---|---|---|---|
| 1st | JFH-1 E2-His / TALON beads | 20μg / 2mg | $4.8 \times 10^{12}$ | 5 | $2.3 \times 10^{8}$ | $1 / (2.1 \times 10^{4})$ |
| 2nd | JFH-1 E2-His / TALON beads | 10μg / 2mg | $2.6 \times 10^{12}$ | 8 | $1.9 \times 10^{9}$ | $1 / (1.3 \times 10^{3})$ |
| 3rd | JFH-1 E2-His / anti-His / Protein G-beads | 10μg / 5μg / 0.75mg | $4.4 \times 10^{12}$ | 8 | $8.1 \times 10^{8}$ | $1 / (5.4 \times 10^{3})$ |
| 4th | TH E2-His / TALON beads | 20μg / 2mg | $3.4 \times 10^{12}$ | 8 | $2.6 \times 10^{9}$ | $1 / (1.3 \times 10^{3})$ |

[a]PFU; plaque forming units (a measure of the quantity of phages that are capable of forming a plaque)

in this study were derived from Japanese liver cancer patients, it was thought thatanit-E2 antibodies could be obtained efficiently. On the other hand, it could only produce antibodies specific to genotype 1b and 2a. Using this screen, we obtained 96 phage clones (e2d001-096) exhibiting specific binding affinity for genotype-1b and -2a HCV E2 protein-adsorbed beads (Table 1).

We cloned and constructed IgG antibody-encoding genes from these phagemid DNAs, and screened the expressed IgGs by enzyme immune assay (EIA) and HCV pseudoparticles (HCVpp) neutralization assay (S2 Fig). We selected 3 antibody-encoding clones (designated e2d066, e2d073, and e2d081) for further characterization, based on the observation that these clones possessed both specific binding affinity for the E2 protein and neutralizing activity against HCVpp infection. Based on the results of EIA, these 3 clones had roughly the same binding affinity for the TH E2 protein (genotype 1b). On the other hand, e2d073 had lower binding affinity for the J6CF E2 protein (genotype 2a) than the other two clones. HCVpp neutralization was assessed using both TH HCVpp and J6CF HCVpp (Fig 1).

In this assay, neutralizing activities for TH HCVpp and J6CF HCVpp were observed with all three IgG-encoding clones. S3 Table shows the 50% infection inhibiting concentration

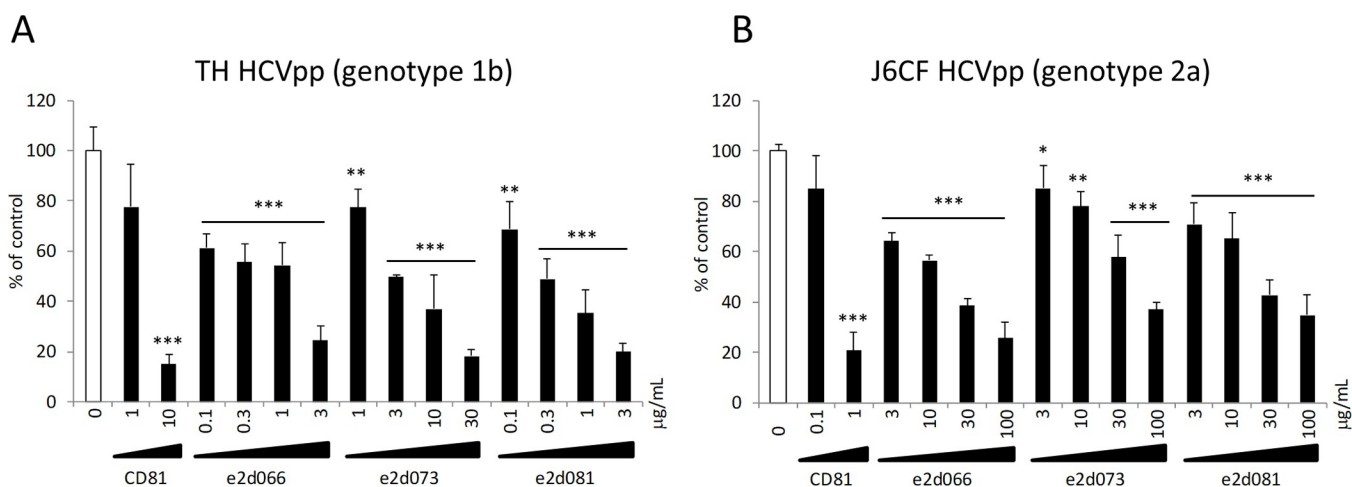

**Fig 1. Neutralizing activities of phage library-derived IgGs against HCVpp infection.** The neutralization effects of the e2d066, e2d073, and e2d081 antibodies were assessed by infection assays using HCV pseudoparticles (HCVpp) harboring the E1 and E2 glycoproteins of (A) TH (genotype 1b) and (B) J6CF (genotype 2a). HCVpp, which include a luciferase-encoding construct, were mixed with immunoglobulin G (IgG) or phosphate buffered saline (PBS) for 30 minutes at room temperature; the mixtures then were used to inoculate naïve Huh-7.5.1 cells. Monoclonal anti-CD81 antibody (JS-81, BD Pharmingen) was used as positive control in this assay. At 72 hours after infection, the infected cells were harvested and lysed with Cell Culture Lysis Reagent (Promega). Luciferase activity was quantified using the Luciferase Assay System (Promega). Neutralizing activity was calculated and is presented as the % neutralization by comparison with the luciferase activities of the well inoculated with the HCVpp-PBS mixture. Assay were performed in triplicate and infection rate are expressed as mean ± SEM. Statistical significance of difference was analyzed using one-way ANOVA with a Williams' test (*P < 0.025, **P < 0.005, ***P < 0.0005 vs PBS).

(IC50) values of these IgG-encoding clones against HCVpp infection. The results revealed that these IgG clones exhibited higher neutralizing activity against TH HCVpp infection than against J6CF HCVpp infection. These data were comparable to the binding affinity to E2 protein as assessed by EIA (S3 Fig). We further analyzed the predicted amino acid sequences of the $V_H$ and $V_L$ regions of the IgGs encoded by these clones; the complementarity-determining regions (CDRs) encoded by the e2d066, e2d073 and e2d081 phages are shown in S4 Table. Notably, e2d066 and e2d081 shared the same CDR in the encode $V_H$ chain. Analysis with igBlast (https://www.ncbi.nlm.nih.gov/igblast/) revealed that these three antibodies were derived from germline VH1-69, which is well known to encode HCV-neutralizing antibodies.

## Epitope analysis with linear epitopes

To identify the epitopes recognized by these antibodies, a library of linear peptides was used. This library spanned the protein sequence of the E2 protein of the TH strain from amino acid (aa) 384 to 717 (where the initiating methionine at the N-terminus of the TH strain precursor protein was defined as aa 1), and consisted of a series of 12-mer peptides (peptides No. 1 to 82) staggered by 3 residues. Each peptide was N-terminally biotinylated, and harbored a C-terminal glycine amide (Fig 2).

Our results showed that the MBL-HCV1 antibody bound to peptide No. 9 (aa 412 to 423), matching the previously reported epitope for this antibody [7] (Fig 2D). In contrast, none of the three clones identified in the present study (e2d066 IgG, e2d073 IgG, and e2d081 IgG) exhibited binding to any members of the peptide library (Fig 2A–2C). This result indicated that the MBL-HCV1 antibody recognizes a linear epitope, while the e2d066 IgG, the e2d073 IgG, and the e2d081 IgG may recognize a conformational epitope.

## Binding of biotinylated antibody to naïve and denatured genotype-1b envelope E2 protein

To determine whether each of the IgG antibodies (e2d066 IgG, the e2d073 IgG, and e2d081 IgG) recognizes a conformational epitope, the antibodies were tested for their ability to recognize denatured E2 protein (Fig 3).

As controls, the assay included the MBL-HCV1 antibody [7], which recognizes a linear epitope of the HCV E2 protein, and the AR3A antibody [21], which recognizes a conformational epitope of the HCV E2 protein. Neither the e2d066 IgG, the e2d073 IgG, the e2d081 IgG, nor the AR3A antibody bound to the denatured TH E2-Fc protein (Fig 3B), while the MBL-HCV1 antibody reacted with both the naïve and denatured TH E2-Fc proteins. These results suggested that the e2d066 IgG, the e2d073 IgG and the e2d081 IgG recognize a conformational epitope, unlike the MBL-HCV1 antibody.

## Comparison of epitopes by competitive EIA

Competitive EIA was used to assess whether the e2d066 IgG recognizes a conformational epitope identical to (or overlapping) that of the e2d073 IgG, the e2d081 IgG, and/or the AR3A antibody (Fig 4).

As control antibodies, the assay included the AR3A neutralizing antibody (which recognizes a conformational epitope), the MBL-HCV1 neutralizing antibody (which recognizes a linear epitope consisting of aa 412 to 423), and the 8D10-3 non-neutralizing antibody (unpublished; which recognizes a linear epitope consisting of aa 522 to 534).

The binding of biotinylated e2d066 IgG to the TH E2-Fc protein was inhibited by the e2d066 IgG and the e2d081 IgG, with the effect depending on the concentration of the competing antibody. Thus, it appears that the e2d066 IgG and the e2d081 IgG recognize the same or

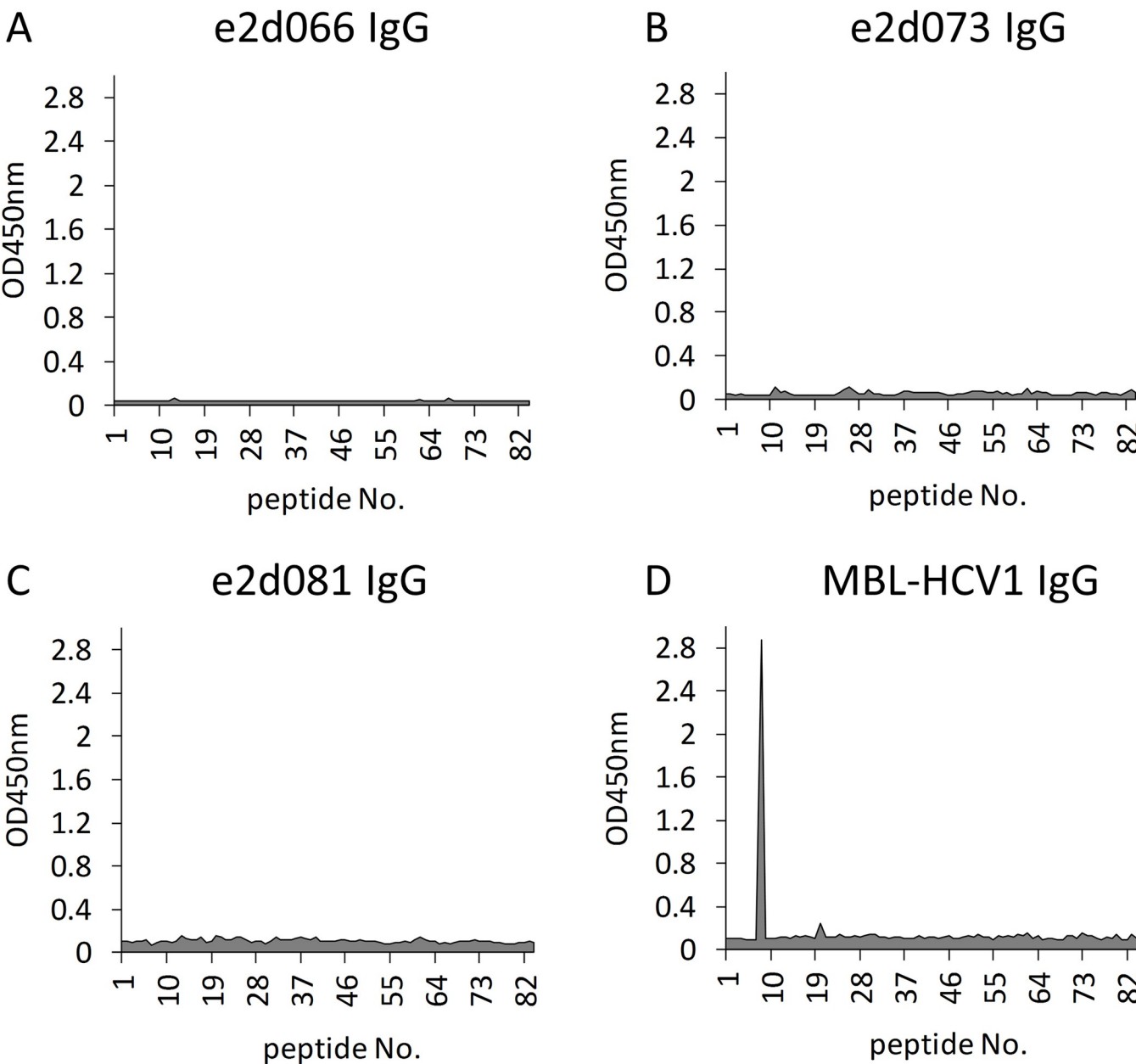

**Fig 2. Epitope analysis with overlapping peptides.** A continuous series of TH E2 12-mer peptides (peptide Nos. 1 to 82), staggered by 3 amino acids each, were synthesized. Each peptide was modified to carry a biotinylation at the N-terminus and a glycine amide at the C-terminus. These peptides were used to coat a streptavidin-coated plate. (A) e2d066, (B) e2d073, (C) e2d081, or (D) MBL-HCV1 antibodies were applied to the peptide-coated plates. HRP-conjugated goat anti-human IgG antibody (Thermo Fisher Scientific) was used to detect the peptide-bound antibodies. The titers of bound antibodies were measured using a peroxidase assay kit (Type T; Sumitomo Bakelite Co., Ltd., Tokyo, Japan).

overlapping epitopes. On the other hand, the e2d073 IgG, the AR3A antibody, the MBL-HCV1 antibody, and the 8D10-3 antibody did not inhibit the binding of the biotinylated e2d066 IgG to the TH E2-Fc protein (Fig 4A), demonstrating that these antibodies recognize a epitope distinct from that bound by the e2d066 IgG.

Next, this competitive EIA was used to assess whether the e2d066 IgG recognizes a conformational epitope identical to (or overlapping) that of the antibody HC-84.1 [22] (Fig 4B).

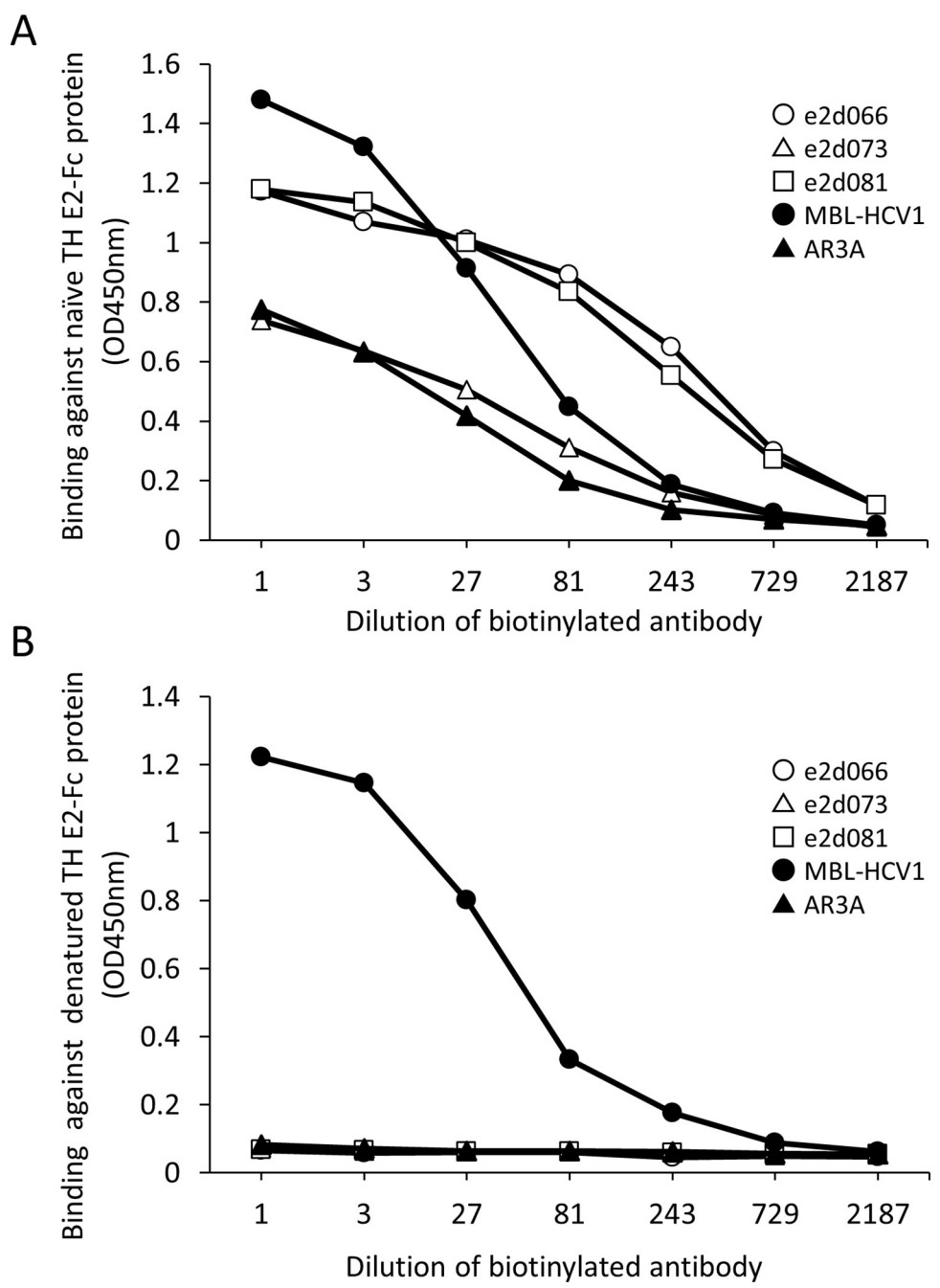

**Fig 3. Antibody binding against naïve and denatured genotype 1b envelope E2 protein.** (A) Naïve and (B) denatured recombinant THE2-Fc proteins were used to coat Nunc-Immune plates (Thermo Fisher Scientific). Biotinylated anti-E2 antibodies were distributed to the prepared plates. Binding of biotinylated anti-E2 antibodies to the coated plates then was assessed using Streptavidin-HRP (GE Healthcare). The titers of antibodies were measured using a peroxidase assay kit for ELISA (Sumitomo Bakelite Co., Ltd.). Open-circle: e2d066 IgG, open-triangle: e2d073 IgG, open-square: e2d081 IgG, filled-circle: MBL-HCV1 (linear epitope control antibody), filled-triangle: AR3A (conformational epitope control antibody).

Notably, HC-84.1 has been suggested to be resistant against the escape mutant emergence. As above, the binding of biotinylated e2d066 IgG to the TH E2-Fc protein was inhibited by the e2d066 IgG and the e2d081 IgG in a dose-dependent manner. In contrast, the HC-84.1, AR3A,

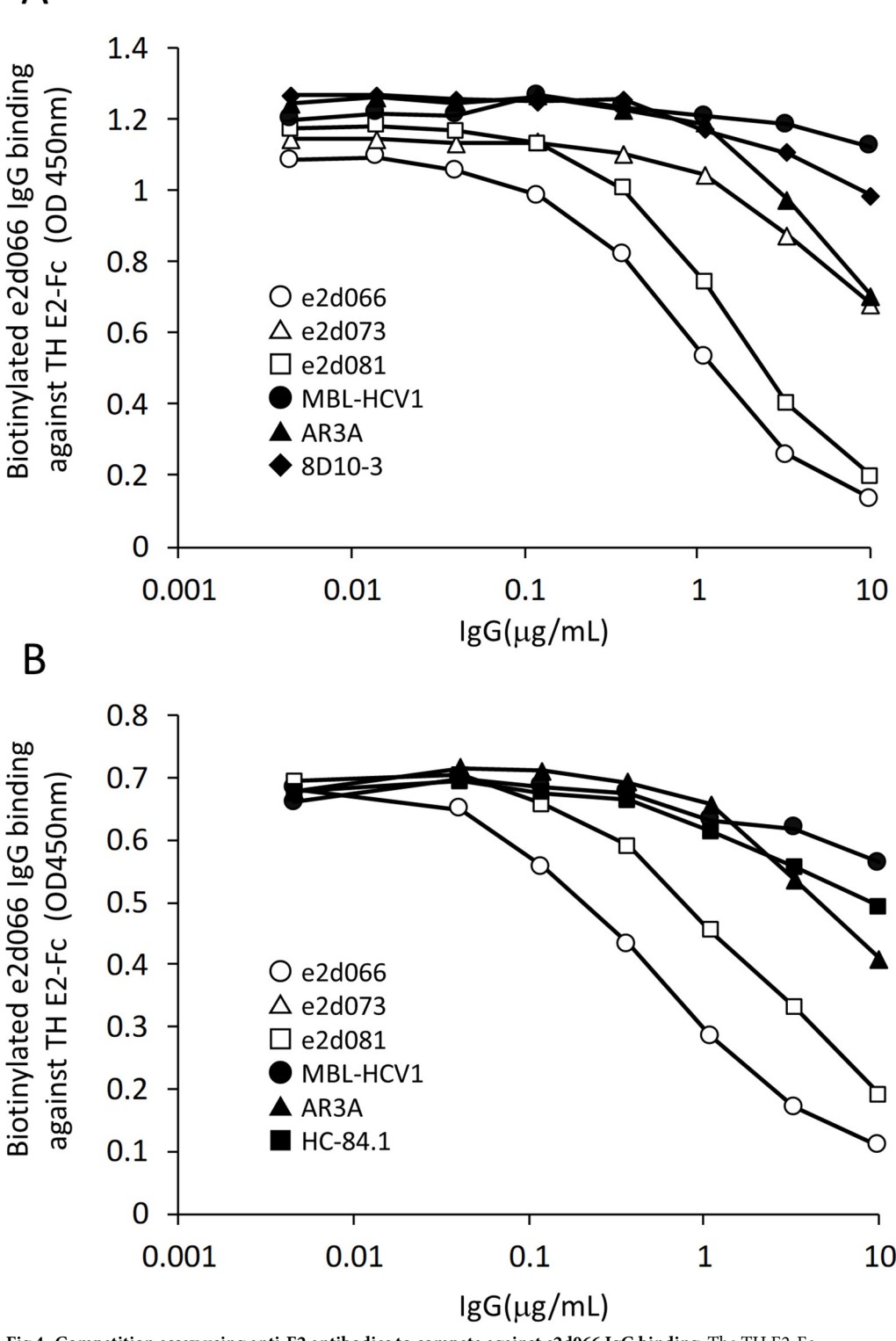

**Fig 4. Competition assay using anti-E2 antibodies to compete against e2d066 IgG binding.** The TH E2-Fc recombinant E2 protein was used to coat Nunc-Immuno plates (Thermo Fisher Scientific). Biotinylated e2d066 IgG was mixed with each of the other anti-E2 antibodies, and the resulting mixtures were distributed to the prepared plates. Avidin-HRP (GE Healthcare) was used to detect the biotinylated IgGs. The titers of antibodies were measured using a peroxidase assay kit for ELISA (Sumitomo Bakelite Co., Ltd.). (A) Competition with linear epitope antibodies and

AR3A. (B) Competition with the escape mutant-suppressive HC84-1 antibody. Open-circle: e2d066 IgG, open-triangle: e2d073 IgG, open-square: e2d081 IgG, filled-circle: MBL-HCV1, filled-triangle: AR3A, filled-rhombus: 8D10-3, filled-square: HC-84.1.

and MBL-HCV1 antibodies did not inhibit the binding of biotinylated ed2066 IgG to the TH E2-Fc protein, demonstrating that the HC-84.1 antibody recognizes an epitope distinct from that of the e2e066 IgG.

## Comparison of the binding of various antibodies to E2 protein

The e2d066 IgG, e2d081 IgG, HC-84.1, MBL-HCV1, and AR3A antibodies were compared for their avidities for the J6CF E2-Fc and the TH E2-Fc proteins (S4 Fig). The results demonstrated that the e2d066 IgG and the e2d081 IgG exhibit stronger binding to the E2 proteins of the J6CF strain (genotype 2a) and the TH strain (genotype 1b) than do the HC-84.1, the MBL-HCV1, and the AR3A antibodies.

## Neutralizing activities of anti-E2 antibody and scFv against HCVcc infection

Next, we assessed the neutralizing activity of the three IgG-encoding clones (e2d066, e2d073, and e2d081) and the corresponding phage-displayed scFv molecules (E2d066scFv, E2d073scFv, and E2d081scFv, respectively) against HCVcc infection. MBL-HCV1, AR3A, and HC.84-1, known human anti-E2 antibodies, also were assessed to permit comparison of inhibition properties and genotypic selectivities. Antibodies were mixed with 100 focus-forming units (FFU) of each of four genotype-chimeric HCVcc (H77/JFH-1(GT1a) [23], TH/JFH-1 (GT1b) [24], J6/JFH-1(GT2a) [14], S310/JFH-1(GT3a) [25, 26]), and inoculated into cultures of Huh-7.5.1 cells. In this assay, HCVcc infection was assessed by measuring the amounts of HCV core protein in the lysates of infected cell; this technique facilitated the processing of a large number of samples. % neutralization was calculated by comparing with the HCV core protein levels in wells inoculated with PBS-treated HCVcc. The infection-neutralizing activities of the IgG and scFv antibodies against HCVcc infection are shown in Fig 5.

These results revealed that both the IgG and the scFv antibodies obtained in the present study have strong neutralizing activity against the chimeric HCVcc of multiple genotypes. Interestingly, the neutralization activities of these IgGs against HCVcc infection were higher than those against HCVpp infection. The IC50 values of the IgG and scFv molecules against HCVcc infection are shown in Table 2.

Notably, the e2d066 and e2d081 IgGs, as well as the corresponding scFv molecules, exhibited neutralizing activity against HCVcc of each of the tested genotypes (1a, 1b, 2a, and 3a). The neutralizing activities of AR3A and HC-84.1 were similar to those of e2d066 and e2d081.

## Evaluation of escape mutant emergence when faced with neutralizing IgG antibodies

We assessed the escape mutant emergence of J6/JFH-1 HCVcc when faced with the IgG antibodies e2d066, e2d073, e2d081, MBL-HCV1, and HC-84.1; an unrelated antibody, HR1-007, was included as a control. The evaluation for escape mutant emergence was performed as shown in S5 Fig. After 8 rounds of screening, we collected the resulting J6/JFH-1 HCVcc-containing supernatants (designated as J6/JFH-1_EM, e.g., J6/JFH-1 EM_e2d066) and assessed for the presence of escape mutant viruses. Specifically, to evaluate the level of the escape mutant viruses, the IC50 values of the antibodies against mutant and control viruses were determined using a focus-forming assay (Table 3 and S5 Table).

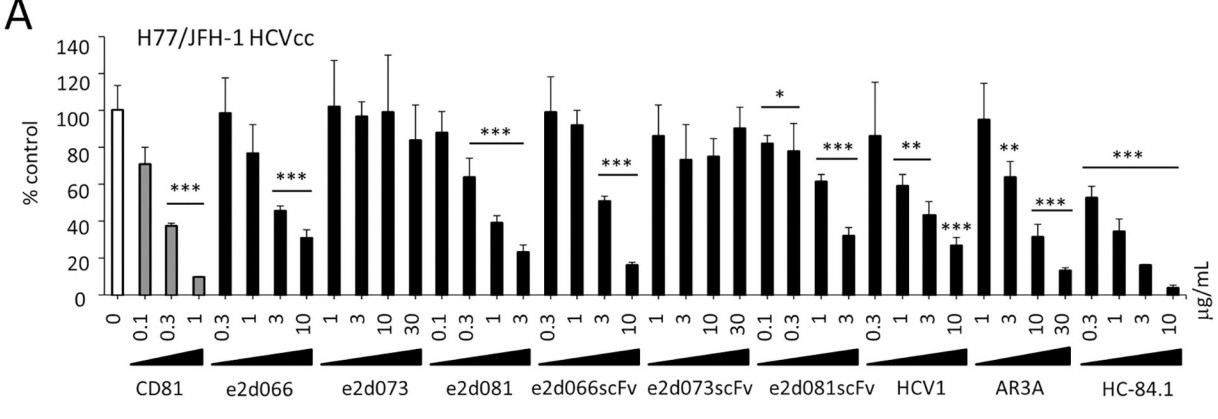

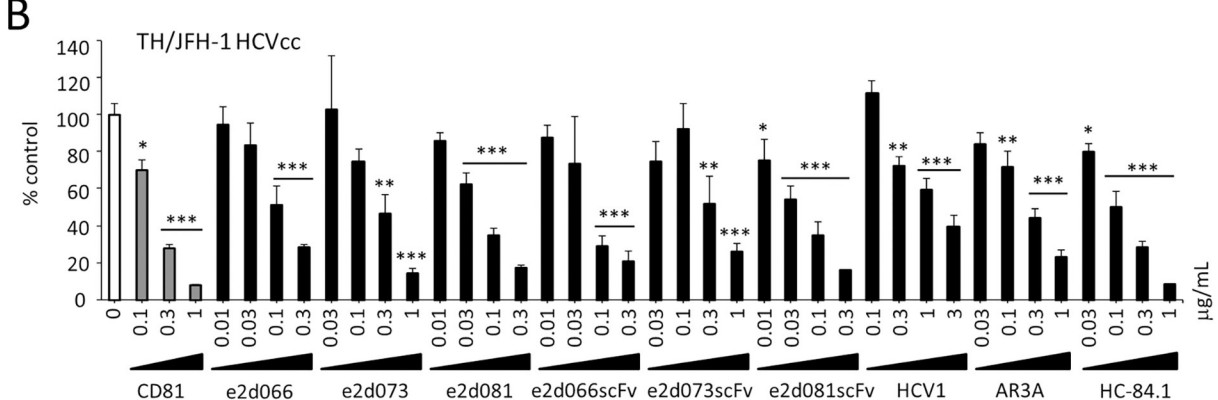

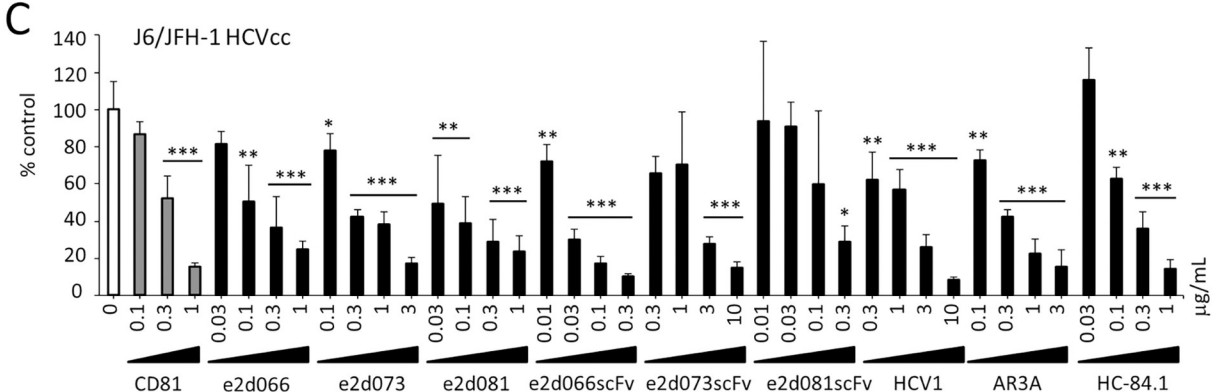

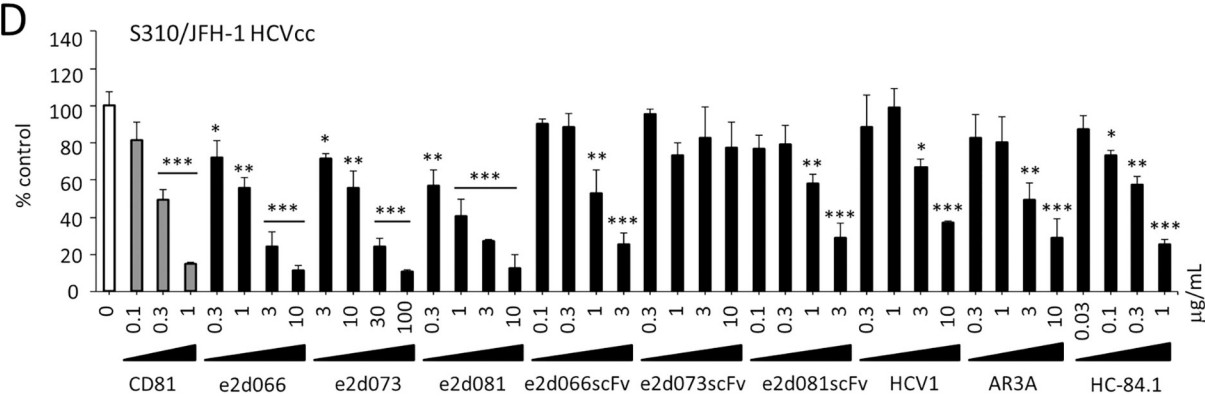

**Fig 5. Neutralizing activity of anti-E2 IgG and scFv antibodies against HCVcc infection.** The neutralization effects of anti-E2 antibodies were assessed using an infection system with chimeric HCVcc, including (A) H77/JFH-1, (B) TH/JFH-1, (C) J6/JFH-1 and (D) S310/JFH-1. A total of 200 focus forming unit of these viruses were mixed with IgG or scFv at the designated concentrations and the mixtures were incubated at room temperature for 30 minutes. The antibody-HCVcc mixture was added to naïve Huh-7.5.1 cells (2 x $10^4$ cells/well, 48-well plate) at 100 μL/well and plates were incubated in a 5% $CO_2$ incubator at 37˚C for 3 hours. The antibody-HCVcc mixture then was removed and replaced with 500 μL of DMEM containing 10% FBS, and the cells were cultured in a 5% $CO_2$ incubator at 37˚C for a further 72 hours. The cells then were washed with PBS, and Passive Lysis Buffer (Promega) was added at 100 μL/well to generate a cell lysate. HCV core protein in the collected cell lysate was quantified using a Lumipulse Ortho HCV Ag (Ortho Clinical Diagnostics, Tokyo, Japan). The efficiency of neutralization was calculated and presented as the % neutralization normalized to the amount of HCV core protein in the HCVcc-PBS mixture-inoculated cell lysate. Assay were performed in triplicate and infection rate are expressed as mean ± SEM. Statistical significance of difference was analyzed using one-way ANOVA with a Williams' test (*$P < 0.025$, **$P < 0.005$, ***$P < 0.0005$ vs PBS).

J6/JFH-1 EM_MBL-HCV1 was highly resistant to MBL-HCV1 antibody neutralization, even at a concentration of 100 μg/mL; however, this mutant virus remained susceptible to neutralization (IC50 <0.01 μg/mL) by the other antibodies (e2d066, e2d073, and e2d081). In contrast, the other mutant viruses (J6/JFH-1_EM_e2d066, J6/JFH-1_EM_e2d073, J6/JFH-1_EM_e2d081, and J6/JFH-1_EM_control) remained susceptible to neutralization by MBL-HCV1. J6/JFH-1_EM_e2d073 and J6/JFH-1_EM_e2d081 were resistant to e2d073 and e2d081 antibody neutralization, respectively. J6/JFH-1_EM_e2d073 was slightly resistant to neutralization by both e2d066 and e2d081, while J6/JFH-1_EM_e2d081 was slightly resistant to neutralization by both e2d066 and e2d073 (Table 3). Interestingly, mutant virus J6/JFH-1_EM_e2d066 remained highly susceptible to neutralization by e2d066, e2d073, and e2d081. We also obtained escape mutant virus J6/JFH-1_EM_HC-84.1 following selection by HC-84.1, an antibody that recognizes a known structural epitope (S5 Table). These results suggested that selection by the e2d066 antibody strongly prevents escape mutant virus emergence.

Characterization of the escape mutant viruses revealed the presence of mutations in the sequences encoding the E2 protein. The resulting E2 protein substitutions included N417T in J6/JFH-1 EM_MBL-HCV1; T387N and I438F in J6/JFH-1_EM_e2d073; and T387N, L433S, and F442L in J6/JFH-1_EM_e2d081.

### In vivo prophylactic study of e2d066 in uPA/SCID human liver chimeric mice

Finally, we tested the neutralizing activity of e2d066 against J6/JFH-1 HCVcc infection in human liver-transplanted uPA/SCID mice. The study design is shown in S6A Fig. Chimeric mice were inoculated with J6/JFH-1 HCVcc on Days 0, 28, and 42 (using HCV RNA copy

**Table 2. IC50 values of antibodies against HCVcc infection, as determined by HCV core EIA.**

| Antibody | IC50[a] value determined by HCV core EIA (μg/mL) | | | |
|---|---|---|---|---|
| | H77 (genotype 1a) | TH (genotype 1b) | J6 (genotype 2a) | S310 (genotype 3a) |
| Anti-CD81 | 0.22 | 0.27 | 0.24 | 0.20 |
| e2d066 | 3.30 | 0.11 | 0.17 | 1.03 |
| e2d073 | >30 | 0.25 | 0.42 | 10.36 |
| e2d081 | 0.66 | 0.05 | 0.02 | 0.50 |
| E2d066scFv | 3.13 | 0.06 | 0.02 | 1.11 |
| E2d073scFv | >30 | 0.35 | 1.03 | >10 |
| E2d081scFv | 1.38 | 0.04 | 0.23 | 1.14 |
| MBL-HCV1 | 2.20 | 0.74 | 0.86 | 7.26 |
| AR3A | 5.75 | 0.24 | 0.26 | 3.24 |
| HC-84.1 | 0.34 | 0.13 | 0.22 | 0.30 |

[a]IC50; 50% infection inhibiting concentration

**Table 3. IC50 values of antibodies against escape mutant HCVcc.**

| Escape mutant viruses | IC50 (μg/ml) for the Indicated IgG Antibodies[a] | | | |
|---|---|---|---|---|
| | e2d066 | e2d073 | e2d081 | MBL-HCV1 |
| J6/JFH-1 EM_control[b] | <0.01 | 0.09 | <0.01 | 0.16 |
| J6/JFH-1 EM_e2d066 | <0.01 | 0.01 | <0.01 | 0.2 |
| J6/JFH-1 EM_e2d073 | 0.97 | 59.4 | 0.04 | 0.31 |
| J6/JFH-1 EM_e2d081 | 0.70 | 5.52 | 0.67 | 0.13 |
| J6/JFH-1 EM_MBL-HCV1 | <0.01 | <0.01 | <0.01 | >100 |

[a]IC50 values (μg/ml) for each antibody against each escape mutant virus was determined by focus-forming assay.

[b] Unrelated control antibody was used to raise this virus.

numbers of $10^2$, $10^3$, and $10^3$ per mouse, respectively). Inoculated animals were treated with the anti-E2 antibody e2d066 or a control human IgG (HR1-007), administered by intraperitoneal injection at 800 μg/mouse, on Day -1 (i.e., one day before) and Days 1, 4, 8, and 11 after the first (Day-0) challenge with J6/JFH-1 HCVcc. Blood samples were collected once weekly throughout the study and used to determined serum viral load and h-Alb through Day 49. Results of this study are shown in Table 4 and S6B Fig.

Two of three control IgG-treated mice showed detectable levels of HCV RNA after the second or third inoculation with J6/JFH-1 HCVcc. However, HCV RNA was not detectable in the sera of any of the three e2d066-treated mice. Little fluctuation of h-Alb levels was observed during the experiment, and h-Alb levels did not change drastically in any of the mice following inoculation with virus. These results suggested that e2d066 has neutralizing and prophylactic activity against HCV infection in vivo, and that treatment with this IgG does not adversely affect the transplanted human liver.

## Discussion

In recent years, massive viral infections have been occurring all over the world. Antivirals are becoming more important approaches to cure acute and persistent viral infectious diseases, but the development of such therapies often takes a long time. Neutralizing antibodies provide an attractive alternative approach to the development of prophylactic and therapeutic agents, given the establishment of a variety of antibody acquisition techniques (e.g., phage display methods) that facilitate the identification of neutralizing antibodies. In the research field of hepatitis C therapy, development of biopharmaceuticals such as neutralizing antibodies and vaccines has lagged behind that of small molecule-based therapies. There is likely to be a need for neutralizing antibodies to prevent HCV accidental infection among health care workers and to prevent reinfection of patients receiving liver transplants. Civacir [7], which consists of polyclonal antibodies obtained from chronic HCV patient sera, and MBL-HCV1 [27] both are undergoing clinical trials in liver transplantation, but the prophylactic effect of these antibodies has yet to be demonstrated clinically. While Civacir and MBL-HCV1 were confirmed to

**Table 4. Neutralizing effect of e2d066 against J6/JFH-1 HCVcc infection in human liver chimeric mice.**

| Inoculated HCV RNA copies | Infected mice | |
|---|---|---|
| | Control IgG (n = 3) | e2d066 (n = 3) |
| $10^2$ | 0/3 | 0/3 |
| $10^3$ (1st) | 1/3 | 0/3 |
| $10^3$ (2nd) | 2/3 | 0/3 |

exhibit neutralizing activity against a broad range of HCV genotypes in vitro, these antibodies were unable to suppress the emergence of escape mutants [28]. Notably, viruses acquired resistance to MBL-HCV1 antibody neutralization through mutations at amino acids 415 and 417 of the E2 protein, which are known glycosylation sites. The emergence of escape mutants via shifts in glycosylation sites is a major obstacle in the development of neutralizing antibodies as therapeutic and prophylactic agents [29]. In the present study, we used phage display screening to identify three anti-HCV antibodies (e2d066, e2d073, and e2d081) with binding specificity for the HCV E2 envelope glycoprotein. These antibodies demonstrated avidity for E2 proteins derived from viruses of both genotypes 1b and 2a, but did not recognize overlapping E2 epitope peptides or denatured E2 protein. Furthermore, these antibodies, both as IgG and scFv, showed cross-genotypic neutralizing activity in in vitro HCV infection assays with J6/JFH-1 (genotype 2a E2), TH/JFH-1 (genotype 1b E2), H77/JFH-1 (genotype 1a E2), and S310/JFH-1 (genotype 3a E2) chimeric viruses, as has been reported previously with structural epitope-recognizing antibodies. In addition, competition with linear epitope-recognizing antibodies (MBL-HCV1, 8D10-3) and structural epitope-recognizing antibodies (AR3A, HC.84-1) at concentrations of up to 1 μg/mL did not impede the binding of e2d066 to E2 glycoprotein. Notably, the MBL-HCV1, 8D10-3, AR3A, and HC.84-1 antibodies are known to recognize epitopes situated in or near the CD81-binding site of E2 glycoprotein, regions designated epitopes I and II [30, 31]. Specifically, the MBL-HCV1 recognizes epitope I (aa 412–423) [8], AR3A binds epitope II (aa 434–446) and aa 523–535 [21], and HC.84-1 binds epitope II and aa 610–619 [22]; in contrast to these neutralizing antibodies, the non-neutralizing antibody 8D10-3 recognizes only aa 522–534 (unpublished). The e2d066 antibody presumably recognizes epitopes distinct from those of the other reported antibodies, although slight competition with reported antibodies was observed at concentrations of 1 μg/mL or higher (Fig 4). We infer that the structural epitope recognized by e2d066 differs from, but may be sterically close to, those of other reported neutralizing antibodies, nonetheless representing a position that is well conserved among HCV genotypes and critical to HCV infection.

The present study also indicated the e2d066 antibody suppresses the emergence of escape mutant viruses. Specifically, after 8 rounds of passaging J6/JFH-1 HCVcc-infected cells in the presence of the e2d066 antibody, the surviving viruses did not show resistance to neutralization (Table 3). In contrast, we were able to isolate escape mutant viruses when passaging in the presence of the e2d073 and e2d081 antibodies. Both J6/JFH-1_EM_e2d073 and J6/JFH-1_EM_e2d081 encoded E2 protein with a T387N substitution, a change located in hypervariable region 1 (HVR1). J6/JFH-1_EM_e2d073 encoded E2 protein with a I438F substitution, and J6/JFH-1_EM_e2d081 encoded E2 protein with L433S and F442L substitutions; in both of these viruses, the changes are located in or near epitope II. HVR1 is known to interact with scavenger receptor class B type I (SR-BI), which is a cell surface receptor for HCV, thereby promoting HCV infection [32]. Furthermore, HVR1 has been reported to inhibit production of antibodies recognizing well-conserved E2 epitopes due to the sequence variability of HVR1 and the ability of this domain to physically protect E2 epitopes [33], suggesting that HVR1 contributes to the development of chronic HCV infection. Therefore, we hypothesize that the e2d073 and e2d081 antibodies may allow escape mutant emergence due to the sequence variability of HVR1. Furthermore, mutations at E2 amino acid residues 433, 438, and 442 have been reported to be responsible for viral escape from neutralization by conformational epitope-recognizing antibodies [34, 35]. The I438F substitution previously was identified following selection by the HC-11 neutralizing antibody; this mutation was reported to reduce the E2 protein's binding affinity for CD81 [34]. The F442L substitution was identified by genotype-1 HCVpp panels and was reported to be resistant to neutralization by the HC84.22 and HC84.26 antibodies [35]. Therefore, we infer that the e2d073 and e2d081 antibodies recognize epitopes

similar or close to those recognized by other reported antibodies, while e2d066 may recognize a distinct epitope. However, further study will be necessary to explain the mechanisms of competition with other antibodies and the escape mutant suppressive property of e2d066.

Interestingly, the structural-epitope antibodies obtained in the present study retained neutralizing activity against escape mutants that emerged following selection with MBL-HCV1 (Table 3). MBL-HCV1 showed similar neutralizing activity (compared to control virus, J6/JFH-1_EM_control) against the escape mutants (J6/JFH-1_EM_e2d066, J6/JFH-1_EM_e2d073, and J6/JFH-1_EM_e2d081) that emerged following selection by the structural epitope-recognizing antibodies identified in this work. These complementary observations suggest that the neutralizing activity of structural-epitope antibodies is unaffected by mutations that emerge following selection by linear-epitope antibodies, and vice versa. This point may reflect the fact that e2d066 and MBL-HCV1 do not compete for binding against the E2 protein (Fig 4). This result indicates that the epitopes recognized by these antibodies are not adjacent, such that binding of one to E2 protein does not sterically block binding of the second. Such synergistic neutralizing effects previously have been reported between neutralizing antibodies that recognize distinct epitopes [36, 37]. Therefore, the combined use of e2d066 and another neutralizing antibody that recognizes a distinct epitope of the E2 glycoprotein may provide synergistic virus neutralization while suppressing the emergence of escape mutants with either antibody alone. Such a combinatory treatment may be beneficial for the treatment or prevention of HCV infection.

Finally, the present study demonstrated that e2d066 exhibits neutralizing activity against HCV infection in vivo. In the relevant experiment, we first confirmed the infectivity of J6/JFH-1 HCVcc in human liver chimeric mice. Two mice inoculated with HCV ($10^2$ copies/mouse) but left untreated with antibody showed detectable serum levels ($10^5$, $10^6$ copies/mL, respectively) of HCV RNA at 7 days post-inoculation. In contrast, in mice inoculated three times with HCV (at sequential doses of $10^2$, $10^3$, and $10^3$ copies/mouse) and treated with the e2d066 antibody, HCV infection was inhibited in all (3 of 3) treated animals. Control animals inoculated three times but treated with isotype antibody demonstrated HCV infection in 2 of 3 animals. Thus, i.p. treatment with the neutralizing antibody e2d066 was sufficient to inhibit HCV infection in vivo. Anti-E2 antibodies inhibit adhesion and entry of HCV via binding to the E2 envelope glycoprotein, impeding the binding of HCV to cell membrane receptors such as CD81, SR-BI, claudin-1, and occludin [38]. It has been reported that anti-CD81 [39], anti-SR-BI [40], anti-claudin-1 [41], and anti-occludin [42] antibodies inhibit HCV infection in vitro and in vivo; like anti-E2 antibody, these "entry inhibitors" act as neutralizing antibodies for HCV infection. It also has been reported that the combination of entry inhibitors and DAAs shows synergistic effects against HCV infection both in vitro and in vivo [43]. Therefore, the simultaneous use of HCV therapeutics with distinct inhibitory mechanisms is expected to prevent viral infection. The e2d066 antibody identified here has an independent inhibitory mechanism against HCV infection and may be a good candidate for use in a combination therapy with other agents (e.g., DAAs) for the treatment of HCV infection.

In conclusion, antibodies obtained by phage display screening recognized conformational epitopes of the E2 glycoprotein and had pan-genotypic neutralizing activity against HCV infection in vitro. In particular, the e2d066 antibody suppressed escape mutant emergence in vitro and showed neutralizing activity in vivo. The e2d066 antibody may be of use for the prevention or treatment of HCV infection.

## Supporting information

**S1 Fig. Schematic illustrations of phage display screening method.** The hepatitis C virus (HCV) E2 protein derived from either strain TH (genotype 1b) or strain JFH-1 (genotype 2a)

was allowed to bind to the surface of magnetic beads, and the single chain Fv (scFv) phage library then was added to the coated beads. Specifically bound phages were amplified by infection of *E. coli*. These steps were repeated for a total of 4 times to enrich for phages with specific binding to the HCV E2 protein. To confirm the specific binding of scFv phages to HCV E2 protein-bound magnetic beads, the gene encoding the corresponding IgG was constructed from phagemid scFv-encoding sequences by PCR cloning. Specific binding of these antibodies to HCV E2 protein was assessed by enzyme immune assay (EIA) recognition and neutralization of HCV pseudoparticle (HCVpp) infection.
(TIF)

**S2 Fig. Results of enzyme immune assay using obtained 96 IgG clones.** Recombinant TH E2-His E2 protein was used to coat Nunc-Immune plates. The prepared plates were used to detect specific bindings of the obtained 96 IgG clones. Horseradish peroxidase-conjugated anti-human immunoglobulin G (IgG-HRP) was used to detect the binding to human IgG. The titers of antibodies were measured using a peroxidase assay kit for ELISA (Sumitomo Bakelite Co., Ltd.).
(XLSX)

**S3 Fig. Comparison of binding affinity to HCV E2 protein of 3 selected IgG clones.** Binding affinity of obtained anti-E2 antibodies to E2 proteins from (A) TH and (B) J6CF was analyzed by enzyme immune assay. The titers of antibodies were measured by a peroxidase assay kit. Open-circle: e2d066 IgG, open-triangle: e2d073 IgG, open-square: e2d081 IgG, filled-square: HR1-007 IgG (control).
(TIF)

**S4 Fig. Comparative study of IgG binding affinity against E2 protein.** Recombinant E2 proteins (A) TH E2-Fc and (B) J6CF E2-Fc were used to coat Nunc-Immune plates. The prepared plates were used to detect specific bindings of the anti-E2 antibodies (e2d066 IgG, e2d081 IgG, MBL-HCV1, AR3A, and HC-84.1). Horseradish peroxidase-conjugated anti-human immunoglobulin G (IgG-HRP) was used to detect the binding to human IgG. The titers of antibodies were measured using a peroxidase assay kit for ELISA (Sumitomo Bakelite Co., Ltd.). Open-circle: e2d066 IgG, open-square: e2d081 IgG, filled-circle: MBL-HCV1 (linear epitope control antibody), filled-triangle: AR3A (conformational epitope control antibody), filled-square: HC-84.1 (conformational epitope control antibody).
(TIF)

**S5 Fig. Schematic illustration of the method used to isolate anti-E2 antibody neutralization-resistant virus.** Each IgG antibody was diluted with phosphate-buffered saline (PBS) and combined with J6/JFH-1 cell culture-generated infectious hepatitis C virus particles (HCVcc); the mixture was allowed to react at 37°C for 1 hour. The antibody-HCVcc mixture then was added to Huh7 cells seeded in a 12-well plate to permit infection of the cells with HCVcc. Three days after the infection, the cells were subcultured into a 6-well plate, and the culture supernatant was collected at 3 and 6 days after the start of subculturing. An infectious titer measurement was performed to confirm that the collected culture supernatant contained J6/JFH-1 HCVcc. The culture supernatants that contained J6/JFH-1 HCVcc were used for subsequent infection, such that the collected culture supernatant again was mixed with the antibody, and the mixture was added to uninfected Huh7 cells. These steps were repeated for a total of 8 times. The infection-inhibiting activity of the anti-E2 antibody against the virus isolated at the final round (J6/JFH-1_EM) was determined to assess the possible presence of an escape mutant.
(TIF)

**S6 Fig. Dosing with anti-E2 antibody e2d066 does not affect serum concentration of human albumin (h-Alb) in human liver chimeric mice.** (A) Study design of in vivo infection experiment. Human liver-transplanted immunodeficient mice were inoculated with J6/JFH-1 cell culture-generated infectious hepatitis C virus particles (HCVcc) on Days 0, 28, and 42 (inoculated HCV RNA copy numbers were $10^2$, $10^3$, and $10^3$ per mouse respectively; filled triangles). Anti-E2 antibody e2d066 or control human IgG were administered by intraperitoneal injection (800 μg/mouse, i.p.; gray triangles) to the mice on Day -1 (i.e., one day before) and Days 1, 4, 8, 11 after the first (Day-0) challenge with J6/JFH-1 HCVcc. Blood samples of the mice were collected once weekly (open triangle) and processed to obtain serum sample. (B) Human albumin in the blood of the chimeric mice was measured with the Alb-II Kit (Eiken Chemical, Tokyo, Japan). Filled points: control IgG treated mice. Open points: e2d066 treated mice.
(TIF)

**S1 Table. List of antibodies.**
(DOCX)

**S2 Table. List of primers.**
(DOCX)

**S3 Table. IC50 values of antibodies against HCVpp infection.**
(DOCX)

**S4 Table. List of CDRs displayed by the scFv phages.**
(DOCX)

**S5 Table. IC50 value of HC-84.1 antibody against escape mutant HCVcc.**
(DOCX)

## Author Contributions

**Conceptualization:** Hiroshi Yokokawa, Tomokatsu Iwamura, Kazuaki Chayama, Takaji Wakita.

**Data curation:** Yuji Teraoka, Michio Imamura, Noriko Nakamura, Noriyuki Watanabe, Tomoko Date, Hideki Narumi.

**Formal analysis:** Hiroshi Yokokawa, Midori Shinohara, Yuji Teraoka, Michio Imamura.

**Funding acquisition:** Noriko Nakamura, Hideki Aizaki, Hideki Narumi, Kazuaki Chayama, Takaji Wakita.

**Investigation:** Hiroshi Yokokawa, Midori Shinohara, Yuji Teraoka, Michio Imamura, Noriko Nakamura, Noriyuki Watanabe, Tomoko Date.

**Methodology:** Hiroshi Yokokawa, Midori Shinohara, Yuji Teraoka, Michio Imamura, Noriyuki Watanabe, Tomoko Date, Tomokatsu Iwamura.

**Project administration:** Hideki Aizaki, Kazuaki Chayama, Takaji Wakita.

**Resources:** Michio Imamura, Hideki Aizaki, Tomokatsu Iwamura, Hideki Narumi, Kazuaki Chayama, Takaji Wakita.

**Supervision:** Hideki Narumi, Kazuaki Chayama, Takaji Wakita.

**Validation:** Hiroshi Yokokawa.

**Visualization:** Hiroshi Yokokawa, Noriko Nakamura.

**Writing – original draft:** Hiroshi Yokokawa.

**Writing – review & editing:** Hiroshi Yokokawa, Takaji Wakita.

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
