## [Decision Letter · Decision Letter 0]

14 Jun 2022

PONE-D-22-15550Patient-derived monoclonal antibody neutralizes HCV infection in vitro and vivo without generating escape mutantsPLOS ONE

Dear Dr. Yokokawa,

Thank you for submitting your manuscript to PLOS ONE. After careful consideration, we feel that it has merit but does not fully meet PLOS ONE’s publication criteria as it currently stands. Therefore, we invite you to submit a revised version of the manuscript that addresses the points raised during the review process. Please submit your revised manuscript by Jul 29 2022 11:59PM. If you will need more time than this to complete your revisions, please reply to this message or contact the journal office at plosone@plos.org. Please include the following items when submitting your revised manuscript:A rebuttal letter that responds to each point raised by the academic editor and reviewer(s). You should upload this letter as a separate file labeled 'Response to Reviewers'.A marked-up copy of your manuscript that highlights changes made to the original version. You should upload this as a separate file labeled 'Revised Manuscript with Track Changes'.An unmarked version of your revised paper without tracked changes. You should upload this as a separate file labeled 'Manuscript'.If applicable, we recommend that you deposit your laboratory protocols in protocols.io to enhance the reproducibility of your results. Protocols.io assigns your protocol its own identifier (DOI) so that it can be cited independently in the future. For instructions see: https://journals.plos.org/plosone/s/submission-guidelines#loc-laboratory-protocols. Additionally, PLOS ONE offers an option for publishing peer-reviewed Lab Protocol articles, which describe protocols hosted on protocols.io. Read more information on sharing protocols at https://plos.org/protocols?utm_medium=editorial-email&utm_source=authorletters&utm_campaign=protocols.

We look forward to receiving your revised manuscript.

Kind regards,

Mitchell Ho, Ph.D.

Academic Editor

PLOS ONE

Journal Requirements:

"Hiroshi Yokokawa, Noriko Nakamura, Tomokatsu Iwamura, and Hideki Narumi are employees of Toray Industries, Inc.

Midori Shinohara is an employee of Medical & Biological Laboratories Co., Ltd."

We note that you received funding from a commercial source: Toray Industries, Inc. and Medical & Biological Laboratories Co., Ltd.

Reviewers' comments:

Reviewer's Responses to Questions

**Comments to the Author**

1. Is the manuscript technically sound, and do the data support the conclusions?

Reviewer #1: Yes

Reviewer #2: Yes

Reviewer #3: Yes

2. Has the statistical analysis been performed appropriately and rigorously? 

Reviewer #1: No

Reviewer #2: Yes

Reviewer #3: No

3. Have the authors made all data underlying the findings in their manuscript fully available?

Reviewer #1: Yes

Reviewer #2: No

Reviewer #3: No

4. Is the manuscript presented in an intelligible fashion and written in standard English?

Reviewer #1: Yes

Reviewer #2: Yes

Reviewer #3: Yes

5. Review Comments to the Author

Reviewer #1: In this study, Yokokawa H et al., have screened monoclonal antibodies neutralizing HCV infection with phage display and identified 3 antibodies. Subsequently, they validated their neutralizing capacity and showed that they may recognize only conformational epitopes but denatured/linear peptides. Lastly, they showed that e2d066 did not produce escape mutants after long-term treatment in vitro and suppressed infection in vivo.

Although DAA can quite efficiently eliminate HCV, considering its high medical costs and limited access to DAA in the developing countries where HCV infection still frequently occur, it is important effort to develop neutralizing antibody. This paper is well-designed and contains solid experimental evidences of new neutralizing antibodies toward HCV, contributing to the therapeutic development to prevent HCV infection. The manuscript is also well-written and have sufficient information for readers. I have a few comments for this manuscript.

1)There is lack of statistical analysis for this paper. Please use appropriate statistics if applicable.

1) Please discuss why e2d066 can suppress the emergence of escape mutant.

2) in In vivo study, mice were pre-tread with antibody before the infection, which may not be the practical situation in human. This had better be described.

3) Suppe Fig S2 should be S1 in line 191 Page 7.

4) The order of Fig 3A and B must be reversed or result description should be modified to fit the order.

Reviewer #2: This work by Yokokawa et al reported cloning and characterization of human monoclonal antibodies that efficiently neutralize hepatitis c virus (HCV). The authors used an approach of phages display to construct three monoclonal antibodies from HCC patients. They demonstrated that these antibodies recognize conformational rather than linear epitopes of HCV E2 envelope protein and display cross-genotypic neutralizing activity against genotype-1a, -1b, -2a, and -3a HCV in cell culture. No viral escaping mutation was selected during the long treatment of one antibody e2d066. Moreover, they showed that antibody e2d066 inhibits HCV infection in vivo using a human hepatocyte-transplanted mouse model. The work was well performed and the major conclusions are justified by the provided data. The developed anti-HCV antibodies have a potential therapeutic application. The paper was well written and deserves to be published at PLoS ONE. I have a few minor comments as follows.

1. The protocol of the phage display library construction should be described in details. The information of the HCC patient from which the scFv library was derived should be presented.

2. The rationale of using E2 of the TH and JFH-1 HCV strains for the initial bead screening should be described in Results and the pros and cons of this choice should be discussed. Why did the authors switch to J6CF rather than JFH-1 for the EIA, HCVpp and HCVcc assays?

3. I would urge the authors to present the EIA and HCVpp data of all 96 phage clones (e2d001-096) exhibiting specific for genotype-1b and -2a HCV E2 protein-adsorbed beads.

4. Many human anti-HCV neutralizing antibodies recognizing conformational viral epitopes are derived from germline gene VH1-69 or VH7-4-1. Can the authors infer which germline gene these three anti-HCV antibodies are derived from?

Reviewer #3: This article described three patient derived antibodies screened from phage display. The topic here is interesting and important. Those antibodies might be useful and beneficial for patients at the end. The manuscript was well written and straightforward. Below are some suggestions for revisions.

1 On the phage panning part, it will be useful to show the number of phage input and output from 4 rounds panning. It will be interesting to show the summary or analysis of the 96 clones.

2 In figure 3 and 4, the group labels are missing in the graph.

3 In table 3, there are only three mice per group. Is the statistics difference significant? what is the statistics here?

4 The IgGs and scFvs have similar activity in Table 1. Can the authors discuss more on the reasons here?

5 Can the authors measure the binding affinity of those antibodies to the antigens by BLI?

6 It will be useful to draw a cartoon showing the mechanisms of the antibodies’ neutralizing activities by targeting E2.

6. PLOS authors have the option to publish the peer review history of their article (what does this mean?). If published, this will include your full peer review and any attached files.

Reviewer #1: No

Reviewer #2: No

Reviewer #3: No

---

## [Author Response · Author response to Decision Letter 0]

30 Jul 2022

Journal Requirements:

Thank you for your comment. We have changed our paper to PLOS ONE style.

"Hiroshi Yokokawa, Noriko Nakamura, Tomokatsu Iwamura, and Hideki Narumi are employees of Toray Industries, Inc.

Midori Shinohara is an employee of Medical & Biological Laboratories Co., Ltd."

We note that you received funding from a commercial source: Toray Industries, Inc. and Medical & Biological Laboratories Co., Ltd.

Amended COI included in cover letter. Please confirm. If there is insufficient information, please indicate.

"data not shown" was deleted and information was added (line 830 Page 27).

We have changed our paper to PLOS ONE style.

We were not aware that the paper I cited had been withdrawn. The withdrawn paper and the text have been erased (line 82 Page 3).

Reviewer #1: In this study, Yokokawa H et al., have screened monoclonal antibodies neutralizing HCV infection with phage display and identified 3 antibodies. Subsequently, they validated their neutralizing capacity and showed that they may recognize only conformational epitopes but denatured/linear peptides. Lastly, they showed that e2d066 did not produce escape mutants after long-term treatment in vitro and suppressed infection in vivo.

Although DAA can quite efficiently eliminate HCV, considering its high medical costs and limited access to DAA in the developing countries where HCV infection still frequently occur, it is important effort to develop neutralizing antibody. This paper is well-designed and contains solid experimental evidences of new neutralizing antibodies toward HCV, contributing to the therapeutic development to prevent HCV infection. The manuscript is also well-written and have sufficient information for readers. I have a few comments for this manuscript.

1)There is lack of statistical analysis for this paper. Please use appropriate statistics if applicable.

Thank you for your suggestion. We have performed a statistical analysis of the results of the HCVpp and HCVcc infection inhibition experiments (Fig 1, Fig 3). It was expected that the inhibition of infection by the antibodies was concentration-dependent and monotonicity, we used the one-way analysis of variance with Williams’ test for statistical analysis.

2) Please discuss why e2d066 can suppress the emergence of escape mutant.

Thank you for your suggestion. Although the three antibodies we obtained in this study were highly active in inhibiting infection, only one of them inhibited the development of escape mutants. When we analyzed the escape mutants obtained from the other two antibodies (e2d073, e2d081), we found that they were viruses with mutations in HVR1. That is, these two antibodies are antibodies that recognize HVR1, which may have caused the escape mutant. The results of the competition assay showed that e2d066 did not specifically compete not only with these two antibodies, but also with the antibody that produced the escape mutant, suggesting that e2d066 recognizes a different epitope from these antibodies. However, there may be other reasons besides the epitope for suppressing escape mutants, and we believe that this is all we can discuss from the data we have at present. We have included your point as a consideration for future study.

3) in In vivo study, mice were pre-tread with antibody before the infection, which may not be the practical situation in human. This had better be described.

Certainly, this is a different experimental method from the situation in human clinical trials. The human liver chimera mice used in this experiment are based on immunodeficient mice and are deficient in T cells. Therefore, administration of anti-E2 antibody does not induce cellular immunity by the antibody and cannot eliminate cells with established infection. For this reason, we administered antibodies in advance and inoculated the virus with sufficient antibodies in the body to verify the effect of inhibiting infection in vivo. This experimental method is often used in infection experiments using human liver chimeric mice and is also used, for example, in the following paper. Desombere I, et al. Monoclonal anti-envelope antibody AP33 protects humanized mice against a patient-derived hepatitis C virus challenge. Hepatology. 2016 Apr;63(4):1120-34. PMID: 26710081.

4) Suppe Fig S2 should be S1 in line 191 Page 7.

Thank you for your suggestion. We have corrected the points you pointed out (line 473 Page 15).

5) The order of Fig 3A and B must be reversed or result description should be modified to fit the order.

Thank you for pointing this out. We have corrected the order to match the text (line 572 Page 19).

Reviewer #2: This work by Yokokawa et al reported cloning and characterization of human monoclonal antibodies that efficiently neutralize hepatitis c virus (HCV). The authors used an approach of phages display to construct three monoclonal antibodies from HCC patients. They demonstrated that these antibodies recognize conformational rather than linear epitopes of HCV E2 envelope protein and display cross-genotypic neutralizing activity against genotype-1a, -1b, -2a, and -3a HCV in cell culture. No viral escaping mutation was selected during the long treatment of one antibody e2d066. Moreover, they showed that antibody e2d066 inhibits HCV infection in vivo using a human hepatocyte-transplanted mouse model. The work was well performed and the major conclusions are justified by the provided data. The developed anti-HCV antibodies have a potential therapeutic application. The paper was well written and deserves to be published at PLoS ONE. I have a few minor comments as follows.

1. The protocol of the phage display library construction should be described in details. The information of the HCC patient from which the scFv library was derived should be presented.

Thank you for your comment. We have detailed the protocol for phage display library construction in Materials and methods section (line 117 Page 5 and S2 Table). We were also able to confirm that the hepatocellular carcinoma patient sample from which the scFv library was derived from a Japanese chronic hepatitis C patient, and HCV genotypes were 1b or 2a, but we were unable to obtain any further information. However, this study was approved by the Medical & Biological Laboratories Co., Ltd Ethics Committee and was implemented according to the Ethical Guidelines for Human Genome/Gene Research enacted by the Japanese Government and the Helsinki Declaration. This information has been added to the Materials and methods section.

2. The rationale of using E2 of the TH and JFH-1 HCV strains for the initial bead screening should be described in Results and the pros and cons of this choice should be discussed. Why did the authors switch to J6CF rather than JFH-1 for the EIA, HCVpp and HCVcc assays?

For the beads, we used E2 proteins of genotype 1b and genotype 2a, which are the most frequently infected genotypes in Japan. Because the mRNA samples used in this study were derived from Japanese liver cancer patients, it was thought that the use of genotype 1b and genotype 2a E2 proteins might allow for more efficient antibody acquisition. On the other hand, E2 proteins of other genotypes could not be recognized, and only antibodies specific to genotypes 1b and 2a could be obtained. This information has been added to the Results section (line 473 Page 15).

The reason for using the J6CF strain instead of the JFH-1 strain in the infection inhibition experiments was to make the conditions similar to those of other chimeric viruses. When preparing the phage library, we selected JFH-1, a representative genotype 2a virus, for screening. However, when comparing the results of HCVcc infection inhibition experiments with other genotype viruses, we thought that the infection inhibition effect could be evaluated under the same conditions by using chimeric viruses for genotype 2a viruses as well as other genotype viruses. Since we used J6CF for the infection inhibition experiment of HCVcc, we also used J6CF for HCVpp as well, so that we could compare the results.

3. I would urge the authors to present the EIA and HCVpp data of all 96 phage clones (e2d001-096) exhibiting specific for genotype-1b and -2a HCV E2 protein-adsorbed beads.

Thank you for your suggestion. The results of EIA have been added (S2 Fig). However, regarding the results of HCVpp inhibition assay, we do not actually possess detailed data. At the beginning of the study, we did not believe that this screening would yield neutralizing antibodies. The reason was that the source gene for antibodies was a chronically infected patient, and we thought that there might be virtually no antibodies to inhibit the infection. Therefore, we planned to obtain antibodies that bind to the E2 protein but do not inhibit infection, so-called "non-neutralizing antibodies”, and use them in other experiments (e.g. vaccine design, E2 protein structure analysis). However, when we were able to obtain three clones of antibodies that inhibited infection and found that they had interesting properties, we began to focus on their analysis. For these reasons, we do not have the HCVpp inhibition data to present to you. We apologize for not being able to meet your expectations. However, all other data we obtained is presented in this paper. In the future, when we obtain antibodies by phage display method in the same way, we will analyze all the obtained clones.

4. Many human anti-HCV neutralizing antibodies recognizing conformational viral epitopes are derived from germline gene VH1-69 or VH7-4-1. Can the authors infer which germline gene these three anti-HCV antibodies are derived from?

This phage library was created by cloning from a variety of germlines without limiting the germline. igBlast analysis showed that these three antibodies were derived from VH1-69, which is well known to encode HCV neutralizing antibodies. I have added the above information to the Results section (line 521 Page 17). Thank you for your suggestion.

Reviewer #3: This article described three patient derived antibodies screened from phage display. The topic here is interesting and important. Those antibodies might be useful and beneficial for patients at the end. The manuscript was well written and straightforward. Below are some suggestions for revisions.

1 On the phage panning part, it will be useful to show the number of phage input and output from 4 rounds panning. It will be interesting to show the summary or analysis of the 96 clones.

Thank you for your suggestion, we have added the phage panning data in a table to the text (line 481 Page 16). And also the results of EIA have been added (S2 Fig). However, regarding the results of HCVpp inhibition assay, we responded similarly to reviewer#2’s comment, we do not actually possess detailed data. At the beginning of the study, we did not believe that this screening would yield neutralizing antibodies. The reason was that the source gene for antibodies was a chronically infected patient, and we thought that there might be virtually no antibodies to inhibit the infection. Therefore, we planned to obtain antibodies that bind to the E2 protein but do not inhibit infection, so-called "non-neutralizing antibodies”, and use them in other experiments (e.g. vaccine design, E2 protein structure analysis). However, when we were able to obtain three clones of antibodies that inhibited infection and found that they had interesting properties, we began to focus on their analysis. For these reasons, we do not have the HCVpp inhibition data to present to you. We apologize for not being able to meet your expectations. However, all other data we obtained is presented in this paper. In the future, when we obtain antibodies by phage display method in the same way, we will analyze all the obtained clones.

2 In figure 3 and 4, the group labels are missing in the graph.

Thank you for your suggestion. Group labels have been added in Fig 3 and 4.

3 In table 3, there are only three mice per group. Is the statistics difference significant? what is the statistics here?

Thank you for your comment. In this experiment, anti-E2 antibodies were preadministered to human liver chimeric mice to test whether they could protect against HCV infection. This experiment is also done in the following paper. Desombere I, et al. Monoclonal anti-envelope antibody AP33 protects humanized mice against a patient-derived hepatitis C virus challenge. Hepatology. 2016 Apr;63(4):1120-34. PMID: 26710081. As shown in Fig. 6 of this paper, when HCV infection is established in human liver chimeric mice, the amount of HCV RNA in the plasma quickly rises and reaches a plateau. Therefore, there is a clear difference in plasma HCV RNA levels between mice treated with control antibody and mice treated with anti-E2 antibody. In addition, even in mice treated with the same control antibody, if the infection is established at different times, the plasma HCV RNA levels will differ significantly, resulting in a wide error range and making statistical analysis difficult. Therefore, we decided to show the effect of anti-E2 antibody on the inhibition of infection in human liver chimeric mice, depending on whether infection was established or not. We have also performed similar experiments in the following papers. Akazawa D, et al., Neutralizing antibodies induced by cell culture-derived hepatitis C virus protect against infection in mice. Gastroenterology 2013;145:447-455 e441-444.

4 The IgGs and scFvs have similar activity in Table 1. Can the authors discuss more on the reasons here?

Thank you for your comment. In this study, we used “�g/mL” as the unit of concentration of the antibody; the molecular weight of IgG-type antibody is 150 kDa and that of scFv is 25 kDa, so scFv is more present as the actual molar amount. Initially, we expected scFv to be more effective in inhibiting infection, but as a result, the infection inhibitory activity expressed in �g/mL was almost the same for IgG and scFv. Since scFv can be produced in large quantities in E. coli, it can be produced at a lower cost than IgG antibodies. scFv is a very interesting molecule because it has a big advantage when used for industrial applications such as pharmaceuticals. However, we found that the scFv we obtained had a lower inhibitory titer against HCV infection than IgG-type antibodies. It is very interesting that a small molecule like scFv can inhibit infection as well as IgG antibodies, but in this paper, we avoided discussion of scFv in order to focus on the properties of our IgG antibodies.

5 Can the authors measure the binding affinity of those antibodies to the antigens by BLI?

Does BLI mean 'Biolayer Interferometory'? We are not familiar with this technology, so we referred to the following URL. https://www.protocols.io/view/antibody-characterizations-by-biolayer-interferome-8epv5146nl1b/v1 We understood that a special machine is needed to measure BLI. Unfortunately, we do not possess the measurement equipment for BLI.

6 It will be useful to draw a cartoon showing the mechanisms of the antibodies’ neutralizing activities by targeting E2.

Thank you for your comment. The E2 protein is HCV spike protein, and HCV uses these spike proteins to infect cells. The mechanism of infection inhibition by neutralizing antibodies targeting spike proteins is already well known. We believe that it is difficult to draw a cartoon depiction of how our antibodies bind to the E2 protein and inhibit HCV infection from our data. Please refer to the following literature (Schlotthauer, F. McGregor, J. Drummer, H.E To Include or Occlude: Rational Engineering of HCV Vaccines for Humoral Immunity. Viruses 2021, 13, 805). Fig 1 of this literature shows the binding sites of neutralizing antibodies. These binding sites are thought to be important for HCV infection. Our results show that e2d066 does not specifically compete with other neutralizing antibodies. e2d066 also recognizes a structural epitope, so it is not likely to bind to the binding site of MBL-HCV1 (linear epitope). Based on these facts, e2d066 is predicted to bind across several sites where 'infection-neutralizing antibodies recognizing structural epitopes' bind. The binding mechanism of e2d066 to the E2 protein will be the subject of future research.

---

## [Decision Letter · Decision Letter 1]

25 Aug 2022

Patient-derived monoclonal antibody neutralizes HCV infection in vitro and vivo without generating escape mutants

PONE-D-22-15550R1

Dear Dr. Yokokawa,

We’re pleased to inform you that your manuscript has been judged scientifically suitable for publication and will be formally accepted for publication once it meets all outstanding technical requirements.

Kind regards,

Mitchell Ho, Ph.D.

Academic Editor

PLOS ONE

Additional Editor Comments (optional):

Reviewers' comments:

Reviewer's Responses to Questions

**Comments to the Author**

1. If the authors have adequately addressed your comments raised in a previous round of review and you feel that this manuscript is now acceptable for publication, you may indicate that here to bypass the “Comments to the Author” section, enter your conflict of interest statement in the “Confidential to Editor” section, and submit your "Accept" recommendation.

Reviewer #1: All comments have been addressed

Reviewer #2: All comments have been addressed

Reviewer #3: All comments have been addressed

2. Is the manuscript technically sound, and do the data support the conclusions?

Reviewer #1: (No Response)

Reviewer #2: Yes

Reviewer #3: Yes

3. Has the statistical analysis been performed appropriately and rigorously? 

Reviewer #1: (No Response)

Reviewer #2: Yes

Reviewer #3: Yes

4. Have the authors made all data underlying the findings in their manuscript fully available?

Reviewer #1: (No Response)

Reviewer #2: Yes

Reviewer #3: Yes

5. Is the manuscript presented in an intelligible fashion and written in standard English?

Reviewer #1: (No Response)

Reviewer #2: Yes

Reviewer #3: Yes

6. Review Comments to the Author

Reviewer #1: (No Response)

Reviewer #2: All of my previous comments have been addressed. I recommend this version of manuscript for publication.

Reviewer #3: (No Response)

7. PLOS authors have the option to publish the peer review history of their article (what does this mean?). If published, this will include your full peer review and any attached files.

Reviewer #1: **Yes: **Takahiro Kodama

Reviewer #2: No

Reviewer #3: No

---

## [Editor Report · Acceptance letter]

2 Sep 2022

PONE-D-22-15550R1 

Patient-derived monoclonal antibody neutralizes HCV infection in vitro and vivo without generating escape mutants 

Dear Dr. Yokokawa:

I'm pleased to inform you that your manuscript has been deemed suitable for publication in PLOS ONE. Congratulations! Your manuscript is now with our production department. 

Kind regards, 

on behalf of

Dr. Mitchell Ho 

Academic Editor

PLOS ONE